# 2D ferroelectric narrow-bandgap semiconductor Wurtzite' type $\alpha$-In$_2$Se$_3$ and its silicon-compatible growth

Yuxuan Jiang[1,2,12], Xingkun Ning[3,12], Renhui Liu [1,2,12], Kepeng Song [4,12], Sajjad Ali [5], Haoyue Deng[6], Yizhuo Li[1], Biaohong Huang[1], Jianhang Qiu[1], Xiaofei Zhu[1], Zhen Fan [6], Qiankun Li[7], Chengbing Qin [8,9], Fei Xue [10], Teng Yang [1,2] ✉, Bing Li [1,2], Gang Liu [1,2], Weijin Hu [1,2] ✉, Lain-Jong Li [11] & Zhidong Zhang [1]

2D van der Waals ferroelectrics, particularly $\alpha$-In$_2$Se$_3$, have emerged as an attractive building block for next-generation information storage technologies due to their moderate band gap and robust ferroelectricity stabilized by dipole locking. $\alpha$-In$_2$Se$_3$ can adopt either the distorted zincblende or wurtzite structures; however, the wurtzite phase has yet to be experimentally validated, and its large-scale synthesis poses significant challenges. Here, we report an in-situ transport growth of centimeter-scale wurtzite type $\alpha$-In$_2$Se$_3$ films directly on SiO$_2$ substrates using a process combining pulsed laser deposition and chemical vapor deposition. We demonstrate that it is a narrow bandgap ferroelectric semiconductor, featuring a Curie temperature exceeding 620 K, a tunable bandgap (0.8–1.6 eV) modulated by charged domain walls, and a large optical absorption coefficient of $1.3 \times 10^6$/cm. Moreover, light absorption promotes the dynamic conductance range, linearity, and symmetry of the synapse devices, leading to a high recognition accuracy of 92.3% in a supervised pattern classification task for neuromorphic computing. Our findings demonstrate a ferroelectric polymorphism of In$_2$Se$_3$, highlighting its potential in ferroelectric synapses for neuromorphic computing.

Ferroelectrics, renowned for their electric-field-induced switchable polarization, have found broad applications including nonvolatile memories, sensors, and actuators[1–3]. The rapid development of these electronic devices demands nanoscale ferroelectrics, with thickness reduced to nanometer scale. In this regard, conventional 3D ferroelectrics suffer from the polarization instability with polarization magnitude and ordering temperature decrease notably when the film thickness is scaled down, due to the increased depolarization field and the effects of interfacial dead layer[4]. In contrast, emerging van der Waals (vdW) 2D ferroelectrics feature atomically sharp interfaces free of dangling bonds and weak interlayer interactions, providing a pathway to eliminate interfacial defects and strains, thus opening a new route towards ferroelectricity at the atomic scale[5]. Various fascinating phenomena have emerged at the 2D level, including the enhanced polarization in SnTe monolayer due to the quantum confinement effect[6], the negative piezoelectric effect in CuInP$_2$S$_6$ correlated with a reduction in the vdw gap due to dipole-dipole interactions[7], the sliding/twisting ferroelectrics stemming from charge transfer and redistribution during the interlayer translation or twisting between bilayer or multilayer 2D materials[8–10], and the discovery of FE metals such as WTe$_2$[11]. Among them, $\alpha$-In$_2$Se$_3$ exhibits intercorrelated out-of-plane (OOP) and in-plane (IP) polarizations. This is a consequence of its distinctive covalent bond configuration, which consists of a five-triangle atomic lattice of Se-In-Se-In-Se layers. The displacements of

the central Se atoms prompt a reorganization of covalent bonds, thereby facilitating the simultaneous switching of both OOP and IP polarizations[12–14]. This structural interlocking provides a stable OOP polarization against the depolarization field down to monolayer, distinct from other displacement-type 2D ferroelectrics such as SnTe with a pure IP polarization[6] and $CuInP_2S_6$ with an OOP polarization[15,16]. In fact, due to the large displacement of Se (-100 pm), $\alpha$-$In_2Se_3$ has been recognized as a fractional quantum ferroelectric (FQFE) discussed in the framework of the modern theory of polarization[17]. Furthermore, different interlayer stacking sequences of 2H and 3R have been reported for $In_2Se_3$, which give rise to intriguing properties including the stacking modulated FE domain-wall type, resistive switching behaviors, and ferroelasticity[18,19]. Besides, the semiconducting nature with a moderate band gap of -1.39 eV[20], makes it capable of performing multifunction simultaneously in one compact device, such as FE-Field effect transistor (FET) to integrate logic and memory functions[21], and optoelectronic devices for signal detection and information storage[22].

Despite these merits, it is still challenging to synthesize the large-area 2D $In_2Se_3$ films. Conventional chemical vapor deposition (CVD) has been intensively used for growing $\alpha$-$In_2Se_3$[13,23–25], which can obtain $\alpha$-$In_2Se_3$ flakes with lateral sizes up to -1200 $\mu m$[24], but only on mica substrate. As for silicon substrates, the dimensions of $\alpha$-$In_2Se_3$ nanosheets are merely a few tens of microns[26]. This seriously hinders their applications in large-scale integrations with silicon electronics. Besides, $In_2Se_3$ exhibits robust polymorphisms with diverse crystal structures ($\alpha$, $\beta$, $\gamma$, $\delta$, etc.)[27]. The uneven gas flow and inhomogeneous vapor distribution inherent to the conventional CVD process easily led to the coexistence of multiple phases within the as-grown film since these distinct phases exhibit subtle differences in their formation energies. Such perturbations are closely correlated with the remote transport growth (RTG) in CVD, where solid precursors are typically positioned upstream of the substrate, and they traverse a considerable distance before reaching the substrate surface to react and form the desired film. By reducing the transport distance of precursors to substrates, such as putting $In_2O_3$ directly underneath the substrate, centimeter-sized continuous $\beta$-$In_2Se_3$ films can be grown on a mica substrate, however, following film-transfer and strain-engineering steps are required to convert it into the desired $\alpha$-$In_2Se_3$ phase[23], limiting their compatibility with silicon electronics. Secondly, $\alpha$-$In_2Se_3$ is predicted to exist in both the distorted zincblende (ZB') and wurtzite (WZ') crystal structures, which exhibit degenerate formation energies yet differ significantly in their IP polarization magnitudes (38 $\mu C/cm^2$ vs. 115 $\mu C/cm^2$) because of different atomic configurations[12]. Ultimately, the WZ' type $\alpha$-$In_2Se_3$ phase is predicted to exhibit an extraordinary light absorption coefficient of -$10^5$ cm$^{-1}$ across a broad range of wavelengths[28,29]. The enhanced FE polarization and optical properties render it highly attractive for use in optoelectronic devices, optically controlled non-volatile memories[30,31], as well as photon-stimulated electronic synapses with low power consumption[32]. However, to date, $\alpha$-$In_2Se_3$ has been exclusively observed in the ZB' structure, and the high-polarization WZ' variant remains experimentally unconfirmed. Consequently, new strategies are highly desired to facilitate the reliable and large-scale synthesis of this intriguing phase directly on a silicon substrate.

To address these issues, we devise an in-situ transport growth (ITG) method combining pulsed laser deposition (PLD) and CVD, to synthesize centimeter-size $In_2Se_3$ films directly on $SiO_2$ (or Si) substrate. Our approach utilizes PLD to grow $In_2O_3$ precursor film on $SiO_2$ substrate, followed by an in-situ conversion into $In_2Se_3$ through reacting with Se vapor. The stable and adequate source supply of $In_2O_3$ leads to the growth of continuous $In_2Se_3$ films. The scanning transmission electron microscopy shows that the as-prepared film has a WZ' crystal structure that stacks in a 1T phase. The piezoelectric microscopy and second harmonic generation measurements confirm that WZ' type $\alpha$-$In_2Se_3$ is an intercalated 2D ferroelectric with a high Curie

temperature of larger than 620 K. Importantly, it is a narrow band gap semiconductor with a band gap of only 0.8 eV and with a high light absorption coefficient of $10^6$/cm. Leveraging these merits, we develop a two-terminal synapse device (Pt/WZ'-$In_2Se_3$/Pt) that can achieve long-term potentiation and long-term depression synaptic functions with good linearity and symmetry. This enables a supervised learning ability with a high pattern recognition accuracy of 92.3% under light illumination. Our study reveals a 2D ferroelectric narrow band semiconductor, provides a unique approach for its silicon-compatible growth, and demonstrates its potential in synaptic devices used for neuromorphic computing.

## Results and discussion

### Silicon-compatible large-area synthesis of WZ' type $\alpha$-$In_2Se_3$ film

In conventional CVD where RTG dominates the growing process (Fig. 1a), the inhomogeneous vapor transport often results in the growing of mixed phases such as $\beta$, $\beta'$, and $\alpha$ (Fig. 1b) in the form of micro-size flakes (Fig. 1c and Supplementary Fig. S1). In contrast, the ISG approach utilizes the amorphous $In_2O_3$ precursor film that was directly deposited on $SiO_2$ substrate by PLD (Fig. 1d). The ultra-short source-substrate distance can avoid the gas disturbing efficiently, allows for the in-situ direct conversion of $In_2O_3$ into continuous $In_2Se_3$ film in centimeter-scale in the Se vapor environment (Fig. 1f). The uniform color contrast of the enlarged optical image (Fig. 1g) suggests the homogeneity of the film, featuring a smooth surface with a roughness $R_a$ of -0.5 nm, as measured by atomic force microscopy (Fig. 1h). The composition of the film was checked by Energy dispersive spectroscopy (EDS), which gives an atomic ratio of In and Se close to 2:3 (inset of Fig. 1g and Supplementary Fig. S2), indicating the formation of $In_2Se_3$ phase. XPS analysis also confirms the complete transformation of $In_2O_3$ into $In_2Se_3$, because no peak from O-1$s$ was observed (Supplementary Fig. S3). Additionally, the 2D layered growth of the film was clearly illustrated by the presence of step terraces on the film surface, which exhibited characteristic height of -0.8 nm-the thickness of a single layer of $In_2Se_3$ (inset of Fig. 1h), and by the series (00 L) diffraction peaks in X-ray diffraction pattern (Fig. 1i). To reveal the specific phase structure of 2D-$In_2Se_3$, we performed Raman spectra measurements at nine typical locations across the film (Fig. 1j). Similar feature of these spectra suggests the uniformity of the film. We selected one typical spectrum, subtracted the background signal, and then carried out peak fittings by using the Lorentzian peak shape functions as shown in middle panel of Fig. 1k. Distinct Raman peaks situated at 75 cm$^{-1}$, 95 cm$^{-1}$, 149 cm$^{-1}$, 172 cm$^{-1}$, 196 cm$^{-1}$, 218 cm$^{-1}$, and 241 cm$^{-1}$ were determined accurately, with their peak statics including peak positions and FWHM represented in Fig. S4. Raman spectroscopy reveals information on the vibration modes of atoms which is very sensitive to the crystal structures including lattice constant, bonding length and angles, and so has been extensively utilized for the phase determination of 2D materials such as $In_2Se_3$[20,33]. Nevertheless, due to their similar crystal structures, and sometimes low quality of the samples, controversial results are often reported in the literature. We thus conducted theoretical Raman calculations on these phases by using our custom-developed QR$^2$-code[34] (details see Methods), which has been successfully used to identify typical 2D materials including graphene and transition metal dichalcogenides (TMDs)[35,36]. We considered not only the conventional single-resonant Raman process but also the double-resonant scattering process to get a more accurate result as represented in the upper panel of Fig. 1j. Both WZ' and ZB' phases exhibit remarkable similarity as anticipated. However, a distinct frequency red-shift is observable in the Raman peaks of the WZ' phase relative to the ZB' phase. Specifically, the wavenumber of the $E^2$ mode decreases from -88 cm$^{-1}$ for the ZB' phase to -85 cm$^{-1}$ for the WZ' phase. Moreover, the primary peak of the WZ' phase can be effectively resolved into two distinct peaks corresponding to the $A_1^1$ mode and $E^2$ mode, whereas these modes are overlapping in the ZB' phase. These

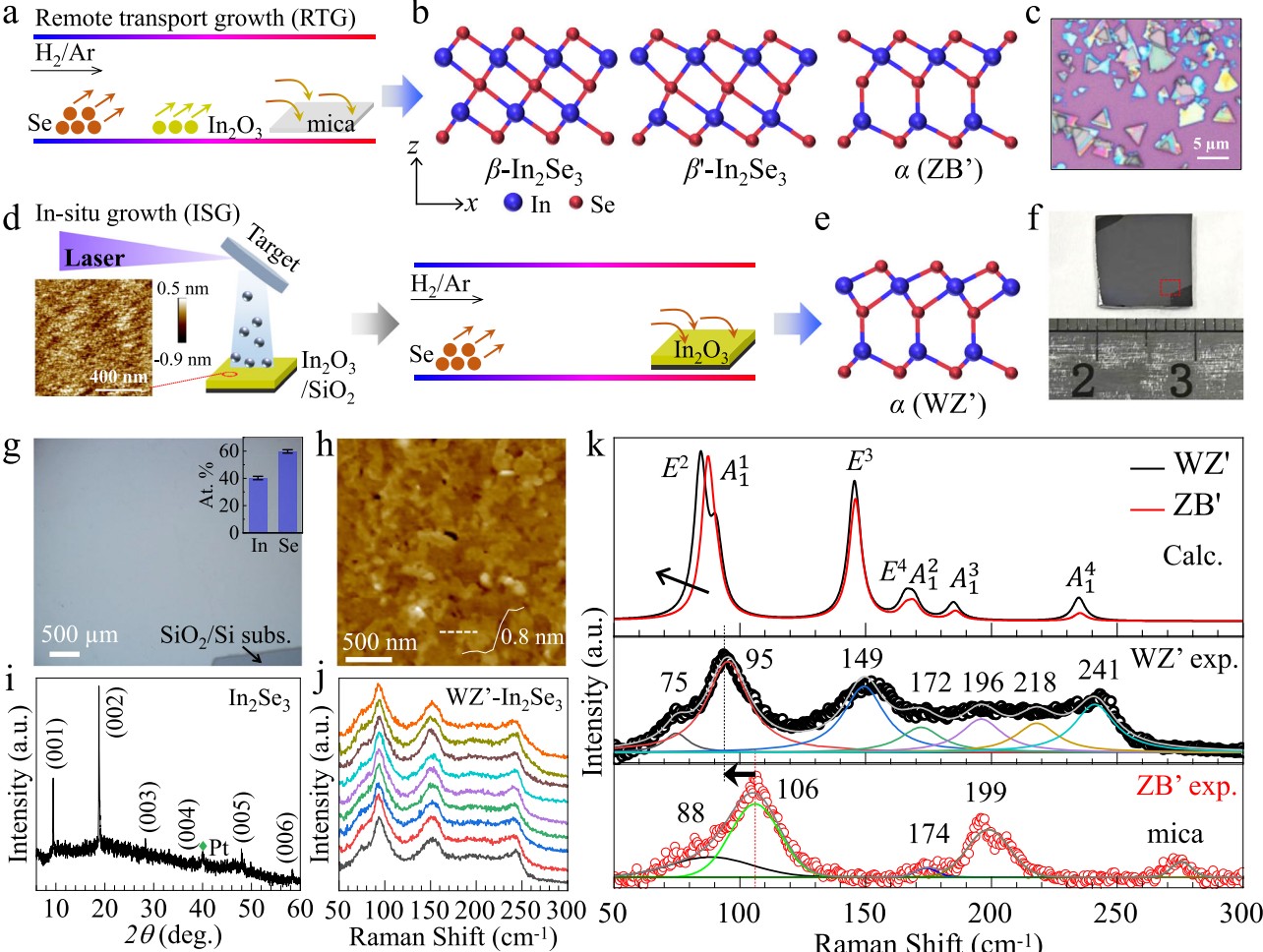

**Fig. 1 | Silicon-compatible large-area preparation of WZ' type α-In₂Se₃ thin films by an in-situ transport growth strategy. a** Schematic of the conventional remote transport growth (RTG) method with an inhomogeneous gaseous precursor distribution. **b** Side-view crystal models of β-, β'-, and α (ZB')-In₂Se₃. The red balls represent Se atoms, and the blue balls represent In atoms. **c** Optical photographs of In₂Se₃ nanosheets by RTG. **d** Schematic of the in-situ transport growth (ITG) method. An amorphous In₂O₃ film was first deposited by PLD, followed by the selenization of In₂O₃ into In₂Se₃ with Se vapor in a CVD furnace. **e** Side-view crystal models of α (WZ')-In₂Se₃. **f** Optical image of a WZ'-In₂Se₃ thin film grown on SiO₂/Si substrate with a size up to 1 cm by 1 cm. **g** The enlarged optical image of the area boxed in (**f**). Inset shows the atomic ratio (-2:3) of In and Se. **h** AFM topography of a randomly selected area on the film. The thickness of a single layer is determined to be -0.8 nm from the line scan shown in the inset. **i** Typical θ−2θ XRD pattern of WZ'-In₂Se₃ film with diffraction peaks indexed as (00 L). **j** Raman spectra acquired at nine different positions. **k** Theoretical and typical experimental Raman spectra of WZ' type and ZB' type α-In₂Se₃ with a thickness of 18 nm.

subtle differences are corroborated by the experimental Raman spectra of the WZ' phase (Fig. 1j, middle panel) and ZB' phase on mica (Fig. 1j, lower panel). The experimental red-shift is up to -11 cm⁻¹, which is roughly four times greater than the theoretical prediction, facilitating the experimental differentiation of the two phases. Additionally, the intensity ratio of $E^3/(E^2+A_1^1)$ is -0.71 in experiment, which aligns closely with the theoretical value of -0.6 (as shown in Fig. S7). This agreement reinforces the dominant formation of WZ' phase in our case. To conclude, the theoretical calculations have captured the dominant red-shift feature of Raman spectra from the ZB' phase to the WZ' phase; however, there are some discrepancies between the calculations and experiments. First, the theoretical Raman peak positions are generally positioned at a smaller wavenumber compared to that of experimental spectra. This is due to the choice of pseudopotentials, specifically generated by the projector augmented wave (PAW) method with the generalized gradient approximation (GGA) for exchange-correlation functional, which typically induces underbinding interatomic potentials and consequently reduces vibrational frequencies. Second, the Raman peaks at -149 cm⁻¹ and -241 cm⁻¹ are nearly absent in the experimental spectra of ZB' phase. These

discrepancies may be due to the thermal excitations that can dampen certain vibration modes at higher temperatures, considering that the theoretical calculations were performed at 0 K. Additionally, interlayer couplings in thicker In₂Se₃ films could also contribute to these differences. Further investigation is required to elucidate the detailed mechanism underlying these observations.

### Ferroelectric semiconducting properties of WZ' type α-In₂Se₃ film

To determine the crystal structure of our film, we performed high-resolution transmission electron microscopy (HRTEM) characterizations on a cross-sectional TEM sample. The low magnification image (Fig. 2a) shows the thickness of In₂Se₃ is -18 nm. Figure 2b represents the typical high-angle annular dark-field (HAADF) image of the film. The atomic resolution allows us to determine the positions of In (marked as blue circle) and Se (denoted by red circle) atoms accurately, yielding an atomic configuration that aligns with the WZ' crystal structure as anticipated theoretically (Fig. 2d). In contrast to the ZB' structure, In atoms in the second layer of WZ' phase are situated above the Se atoms in the fifth layer along the z-axis. As indicated by the red

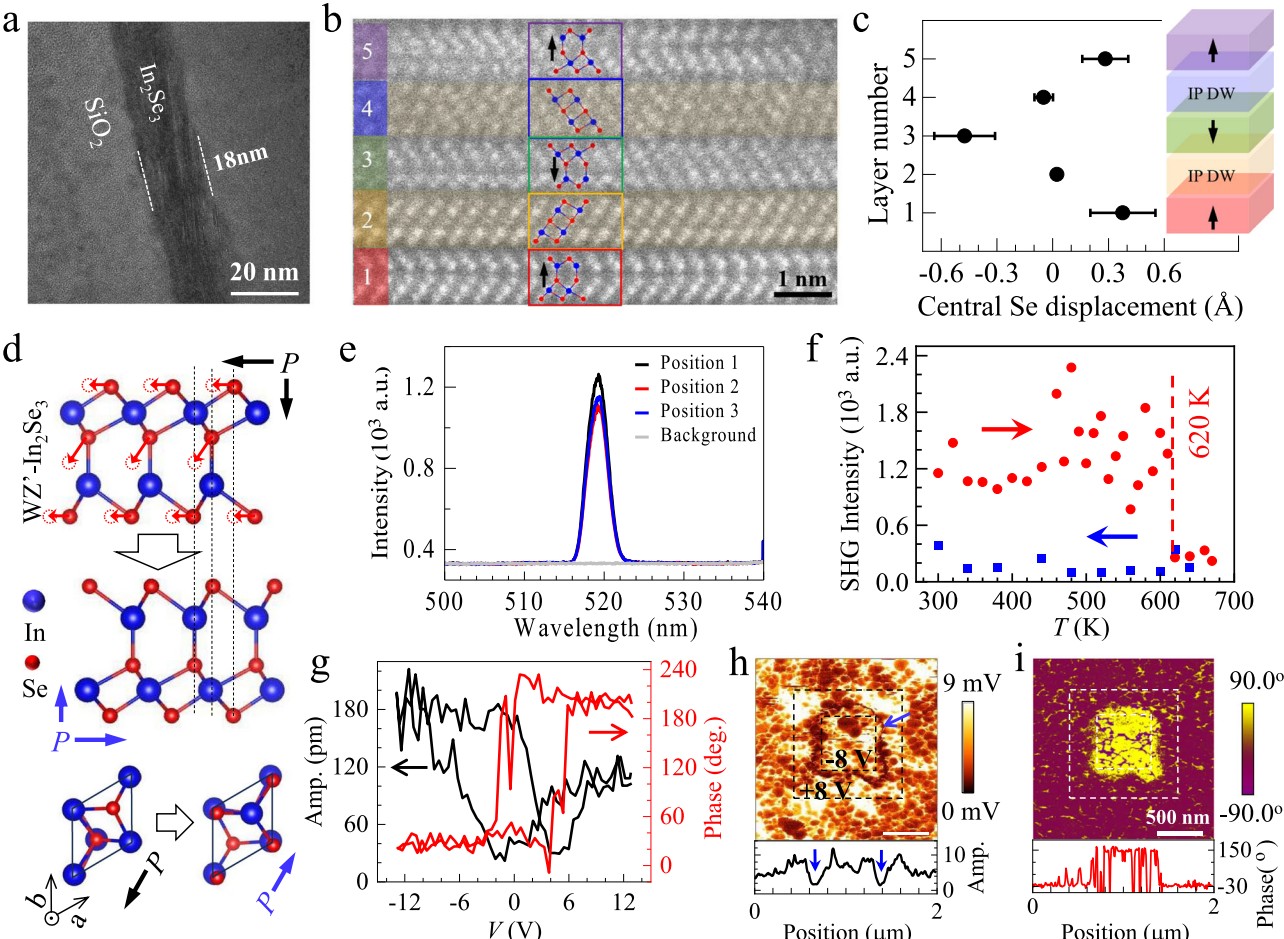

**Fig. 2 | TEM, SHG, and PFM characterizations of WZ' type α-In₂Se₃. a** Low magnification cross-section TEM image. **b** HRSTEM image showing the ferroelectric domain configuration of WZ'-type α-In₂Se₃. WZ' phase of different polarization directions (layers 1, 3, and 5) are isolated by In-plane domain walls (IP-DW, layers 2 and 4). Their atomic microstructures are schematically shown in typical regions marked by rectangles of different colors. **c** The displacement of the central Se atom in each layer along the *c* axis. The error bars represent the standard deviation calculated from 10-unit cells counted per layer. Inset schematically shows the IP-DWs and OOP polarizations. **d** Schematic polarization switching of WZ' phase. The cooperative displacement of Se atoms (indicated by red arrows) induces the simultaneous switching of IP and OOP polarization. IP polarization is along [110]. **e** Room-temperature SHG signal detected randomly at different spots on the film. No SHG signal was observed for the SiO₂ substrate. **f** Temperature-dependent SHG signal between 300 K and 680 K (red circle, warming; blue square, cooling). The SHG signal suddenly decreases at ~620 K as indicated by the dashed line. **g** Local piezoelectric amplitude (amp., left) and phase (right) hysteresis as a function of dc voltage bias sweeping between −12 V and +12 V. **h** Out-of-plane (OOP) PFM amplitude, and **i**, phase images with box domain written by +8 V and −8 V dc voltages (indicated by dashed square lines). The corresponding line scan profiles are shown in the bottom panels. Blue arrows indicate the domain wall positions.

arrows, the polarization reversal is achieved by the concurrent displacement of the top, central, and bottom Se atoms, a process that differentiates it from the ZB' phase where only the central Se atom moves during polarization switching. This scenario is supported by the theoretical prediction of a threefold enhancement in the IP polarization of WZ' phase relative to the ZB' phase (Supplementary Fig. S8). Furthermore, three ferroelectric domains are observed in Fig. 2b (layers 1, 3, and 5), which are separated by in-plane domain walls (IP-DWs) of varying types: layer 2 represents a head-to-head IP-DW, while layer 4 is a tail-to-tail IP-DW. These IP-DWs display a uniform non-polar state, with the central Se atoms positioned in the center of two neighboring In layers. This arrangement marks a sharp flip of the polarization vector direction, with the IP-DW width measuring a single unit cell along the *c*-axis (~0.8 nm)—even narrower than that of traditional ferroelectric oxides like BiFeO₃ (~2 nm[37]). Similar IP-DW has been observed in 2H ZB'-type In₂Se₃[18]. Additionally, we conducted a statistical analysis of the off-center displacements of central Se atoms within the domain to evaluate the magnitude of spontaneous polarization. The displacement, measured as the distance from the central Se atoms to their symmetric position along the *c*-axis, is ~0.3–0.4 Å as illustrated

in Fig. 2c. This aligns well with the theoretical value of ~0.4 Å. By averaging the intensities over a column of atoms, the IP and OOP lattice constants *a* and *c* were determined to be ~0.36 nm and ~0.68 nm, respectively, which are close to the theoretical values of ~0.41 nm and ~0.68 nm[12]. Note that the WZ' type α-In₂Se₃ observed here is different from γ-In₂Se₃, the sole known three-dimensional polymorph of In₂Se₃ that crystallizes in a defective wurtzite structure with vacancy spirals[33,38]. The crystal structure and lattice parameters of γ-In₂Se₃ are totally different from that of WZ' type α-In₂Se₃ (Fig. S9 and Table S1).

To investigate the ordering temperature, we conducted nonlinear optical second-harmonic generation (SHG) measurements on the WZ' type α-In₂Se₃ film. SHG can effectively detect the breaking of inversion symmetry in materials, making it a powerful tool for studying the ferroelectric order and phase transitions[14,16,25]. As shown in Fig. 2e, we observed an intense SHG peak at 519 nm wavelength (with a 1040 nm laser source) at randomly selected positions on the film, whereas no peak was observed for the SiO₂/Si substrate, strongly indicating the existence of ferroelectricity. To further explore the Curie temperature ($T_c$) at which the ferroelectric-paraelectric transition occurs, we

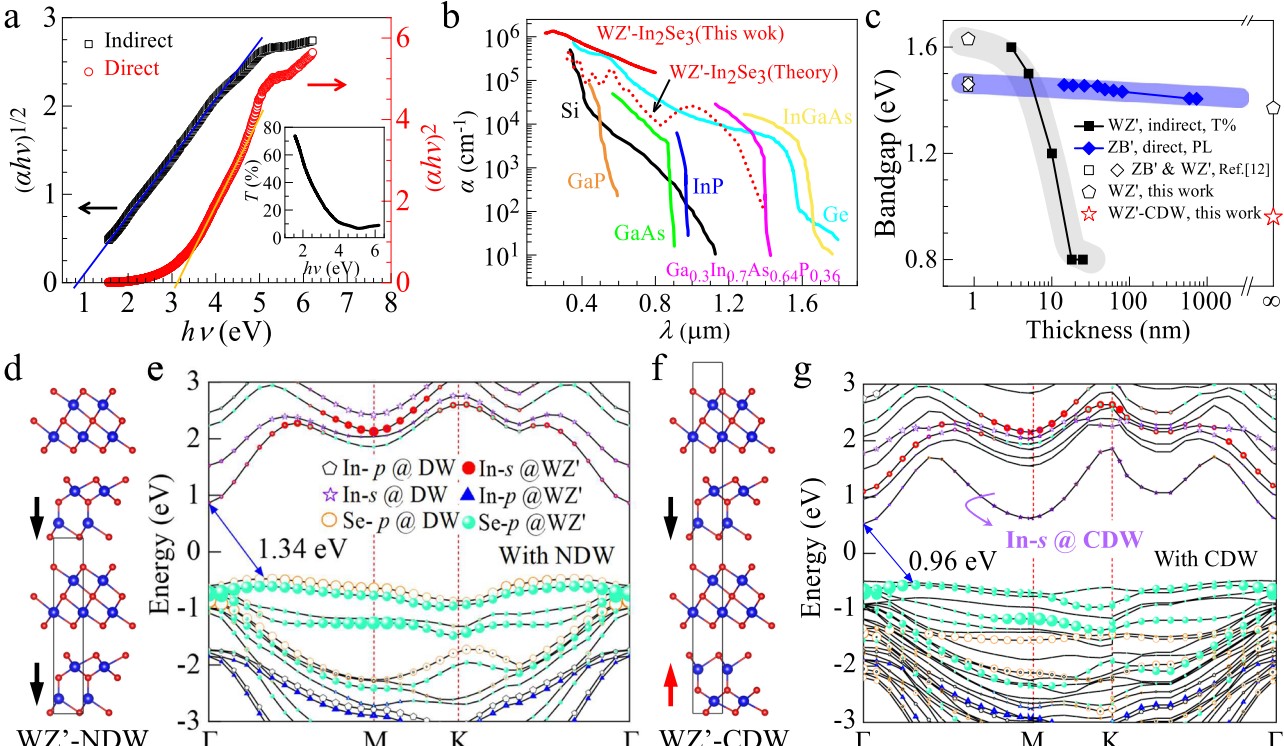

**Fig. 3 | Optical properties of WZ' type $\alpha$-In$_2$Se$_3$. a** $(\alpha h\nu)^{1/2}$ (left) and $(\alpha h\nu)^2$ (right) Tauc plots used for determining the bandgaps for WZ'-type In$_2$Se$_3$ with a thickness of 18 nm. Solid lines are the linear fittings. The inset shows the transmission spectrum within the wavelength range of 200 nm to 800 nm. **b** Optical absorption coefficient of typical semiconductors[45] and WZ' type $\alpha$-In$_2$Se$_3$. **c** The thickness dependent $E_g$ for WZ' type $\alpha$-In$_2$Se$_3$ (this work) and ZB' type $\alpha$-In$_2$Se$_3$[12,30]. Empty symbols represent theoretical values. Thickness of $\infty$ represent bulk. **d** Crystal structure of WZ' phase after including the neutral domain wall (WZ'-NDW), and (**e**) the corresponding projected electronic band structure. **f** Crystal structure of WZ' phase after including the charged domain wall (WZ'-CDW), and (**g**) the corresponding projected electronic band structure. Solid arrows in (**d**) and (**f**) represent the OOP polarization.

performed temperature-dependent SHG measurements across a range of temperatures between 300 K and 680 K (Fig. 2f). We found that the SHG intensity suddenly drops to the noise level at about 620 K during the warming process, which cannot be recovered during the cooling process. The subsequent optical image and Raman spectra (Supplementary Fig. S10) suggest that the sample was partially destroyed and transformed into glassy Se after the characterization[39]. In other words, $T_c$ of WZ' type $\alpha$-In$_2$Se$_3$ is higher than 620 K. We then investigate the polarization switching by piezoelectric force microscopy (PFM). As shown in Fig. 2g, the single-point butterfly-like voltage-dependent piezoelectric amplitude (left) and the 180° phase hysteresis (right) clearly indicate the local ferroelectric polarization switching induced by tip voltage ranging between −12 V and +12 V. To reveal the collected polarization switching, we further acquired the amplitude (Fig. 2h) and phase (Fig. 2i) mappings after writing square patterns with alternative +8 V and -8 V voltages (indicated by dashed lines). The pronounced 180° phase inversion between the -8 V region and the pristine area signifies an upward domain created by the writing voltage, with the domain boundaries exhibiting negligible piezoelectric amplitude, as indicated by the blue arrows. It is worth noting that the application of a +8 V voltage results in a slight increase in the amplitude without changing the phase relative to the pristine area. This suggests a preferred downward self-polarization domain, which can be attributed to the built-in electric field that typically develops when a ferroelectric semiconductor interfaces with a metal electrode[40]. Finally, by investigating the piezoelectric resonant peaks excited by various ac voltages (Fig. S11), we are able to quantitatively determine the piezoelectric coefficient $d_{33}$ of WZ'-type $\alpha$-In$_2$Se$_3$ (18 nm thick), which is found to span from 1.8 pm/V to 4.6 pm/V at seven randomly selected positions on the film. These values are comparable to those of typical 2D

ferroelectrics including ZB' type $\alpha$-In$_2$Se$_3$ (20 nm, $d_{33}$ ~ 2.8 pm/V)[41], 3R-MoS$_2$ (18 nm, $d_{33}$ ~ 0.88 pm/V)[42], and InSe (120 nm, $d_{33}$ ~ 4 pm/V)[43], etc.

One distinct advantage of 2D ferroelectrics is the semiconducting feature with a moderate bandgap (e.g., ZB' type $\alpha$-In$_2$Se$_3$ has a direct $E_g$ of 1.39 eV[20]), which makes them suitable for developing electronic devices integrated with both memory and logic functions. To investigate the semiconducting properties of WZ' type $\alpha$-In$_2$Se$_3$, we conducted optical ultraviolet-visible transmission spectrum measurement on the film with wavelength ranging from 200 nm to 800 nm (inset of Fig. 3a), from which we obtained the absorption coefficient $\alpha$ through $\alpha = -\mathrm{Ln}\,T/d$ with $T$ the transmittance and $d$ the thickness of the film (18 nm) (Fig. 3b). To quantitative determine the bandgap ($E_g$) and its transition type, we applied the empirical Tauc equation, $(\alpha h\nu)^n = A\left(h\nu - E_g\right)$, with $h\nu$ the photon energy, $A$ the proportional constant, and $n$ the transition type ($n = 1/2$ and 2 for indirect- and direct-allowed transitions, respectively)[44]. Extrapolating the linear regions of Tauc plots [$(\alpha h\nu)^n$ vs. $h\nu$, Fig. 3a] yielded an indirect bandgap of ~0.8 eV and an apparent direct bandgap of ~3.1 eV. To resolve this ambiguity, photoluminescence (PL) spectroscopy was performed (Supplementary Fig. S12). Since direct bandgap semiconductors exhibit a distinct PL peak near the band edge due to efficient radiative recombination—a feature absent in our measurements—we conclusively establish bulk WZ'-type $\alpha$-In$_2$Se$_3$ as an indirect semiconductor with $E_g$ of 0.8 eV. The small $E_g$ of WZ' type $\alpha$-In$_2$Se$_3$ makes it capable of absorbing light efficiently as indicated by its large $\alpha$ (~1.3×10$^6$/cm at a wavelength of 244 nm, Fig. 3b) when compared to typical 3D semiconductors including Si, GaAs, and Ge, etc[45]. This is different to the theoretical prediction that WZ' phase has an $E_g$ of ~1.5 eV[12]. To clarify this discrepancy, we investigated the optical properties of WZ' type $\alpha$-In$_2$Se$_3$ films with varying thickness from 25 nm to 3 nm (Fig. S13), and

derived the thickness-dependent $E_g$ as depicted in Fig. 3c. The data of ZB' phase[30] are shown for comparison. We find that the indirect bandgap of 3 nm $In_2Se_3$ is ~1.6 eV, which is in close agreement with the ~1.5 eV predicted for single-layer $\alpha$-$In_2Se_3$ using the HSE06 method[12]. However, the bandgap reduces rapidly to ~0.8 eV as the film thickness exceeds 18 nm, demonstrating a pronounced thickness dependence. This behavior is consistent with observations in various 2D material systems, including TMDCs like $MoS_2$ and $PtSe_2$, which are often attributed to the quantum confinement, interlayer couplings, and dielectric screening effect[46]. To unravel the origin of thickness-dependent $E_g$ in our case, we performed electronic band calculations by using HSE06 (Fig. S14). Three cases have been considered: (i) WZ'-$In_2Se_3$ without domain wall, (ii) bulk WZ'-$In_2Se_3$ with neutral domain wall (NDW), and (iii) bulk WZ'-$In_2Se_3$ with charged domain wall (CDW). We found that CDW effectively reduces the bandgap down to ~0.96 eV (Fig. S14c), obviously smaller than that without domain wall (~1.63–1.37 eV, Fig. S14a) and with NDW (~1.34 eV, Fig. S14b). These results strongly suggest that CDW accounts for this distinct thickness dependence. To elucidate the origin behind, we further performed projected band calculations on bulk WZ'-$In_2Se_3$ with NDW (Fig. 3d, e) and with CDW (Fig. 3f, g). Compared with NDW, where the conducting band minimum is dominated by the s orbitals of In of the WZ' layer (Fig. 3e); we find s orbitals of In atoms in CDW contribute an additional conduction band located closer to the Fermi level, therefore reducing the bandgap efficiently to 0.96 eV. A similar effect has been observed for conventional ferroelectric oxides like $BiFeO_3$, where CDW reduces the bandgap by 0.25-0.5 eV due to the combined effect of defects like oxygen vacancies and the charge screening inside the domain wall[47]. By calculating the charge distribution of WZ'-$In_2Se_3$ with CDW (Fig. S15), we find that charge screening also occurs in CDW of WZ'-$In_2Se_3$. A head-to-head (tail-to-tail) domain will result in CDW with negative (positive) charges. And the long-range attractive Coulombic interactions between neighboring CDW with opposite charges can reduce the total energy from ~−3.632 eV/atom to ~−3.636 eV/atom, making it energetically stable when compared to WZ'-$In_2Se_3$ with NDW (~−3.634 eV/atom). We summarize the key parameters of typical 2D FE semiconductors including $CuInP_2S_6$[16], $In_2Se_3$[20,48,49], $MoS_2$[50], $InSe$[43,51,52], $SnS$[53,54], $SnSe$[55], $SnTe$[6], $MoTe_2$[56], $ReS_2$[57], $CuCrS_2$[58] and $GaSe$[59] in Table 1. The high $T_c$ of exceeding 620 K for WZ'-$In_2Se_3$ ensures the stability of ferroelectric properties at room temperature, which is advantageous for ferroelectric electronic devices to operate stably under ambient conditions. Moreover, the achieved narrow bandgap (down to 0.8 eV) represents, to our knowledge, the smallest experimentally reported value among all 2D ferroelectric semiconductors. Attributes such as high carrier concentration and long-wavelength optical response are typically associated with narrow bandgaps, which promise their potential applications in optoelectronic devices, and photocatalysis, among other fields.

## Electric properties and synapse devices

Now we turn to show the potential applications of WZ' type $\alpha$-$In_2Se_3$ in information storage memories. We constructed the Pt/$In_2Se_3$/Pt in-plane (IP) device on $SiO_2$ insulating substrate (inset of Fig. 4a) and Pt/$In_2Se_3$/Si out-of-plane (OOP) device (inset of Fig. 4g) to take advantage of the IP and OOP polarizations, respectively. For IP devices, the typical thickness of $In_2Se_3$ is 18 nm and the gap between Pt electrodes is 6 μm; for OOP devices, the thickness of $In_2Se_3$ is 200 nm. Figure 4a represents the typical I–V curve of the IP device with a voltage sweeping sequence of -6 V → 6 V → -6 V as indicated by the red arrows. As the voltage amplitude increases, the current increases nonlinearly with two current peaks emerging at -2 V and −2 V, respectively. These current peaks are transient, as shown in Fig. 4b, where they only appear in the first sweeping sequence after setting the device to the opposite polarization state but disappear in the subsequent sweeping period because FE polarization flipping has completed. In addition, as

**Table 1 | Comparison of the key parameters for 2D ferroelectric semiconductors**

| Material | P direction | $E_g$ Type | $E_g$ (eV) | $T_c$ (K) | Ref. |
|---|---|---|---|---|---|
| $CuInP_2S_6$ | OOP | Direct | 2.8 | 320 | 16 |
| ZB' type $\alpha$-$In_2Se_3$ | IP, OOP | Direct | 1.39 | ~700 | 14,20 |
| $\beta'$-$In_2Se_3$ | IP | Indirect | – | 473 | 48 |
| $\gamma$-$In_2Se_3$ | IP, OOP | Direct | 1.95 | – | 20,49 |
| 3R-$MoS_2$ | OOP | Indirect | 1.2 | 650 | 50 |
| $\beta$-InSe | IP, OOP | Direct | 1.28 | – | 51,52 |
| $\gamma$-InSe | IP, OOP | Direct | 1.2 | 300 | 43 |
| SnS | IP | Indirect | 1.23 | – | 53,54 |
| SnSe | IP | Direct | 2.13 | 400 | 55 |
| SnTe | IP | Direct | 1.6 | 270 | 6 |
| d1T-$MoTe_2$ | OOP | Direct | 1.2 | 330 | 56 |
| 1T'-$ReS_2$ | OOP | Direct | – | 405 | 57 |
| $CuCrS_2$ | IP, OOP | Indirect | – | ~700 | 58 |
| $\gamma$-GaSe | IP, OOP | Direct | 2.01 | – | 59 |
| WZ' type $\alpha$-$In_2Se_3$ | IP, OOP | Indirect | 0.8 (bulk) 1.6 (1 L) | >620 | This work |

presented in Fig. 4c, the switching voltages ($V_c$) are frequency dependent, i.e., they shift to larger values as the sweeping frequency increases, obeying a power law $V_c \sim f^\beta$ as predicted from domain-wall motion limited Kolmogorov–Avrami–Ishibashi (KAI) model for ferroelectric switching[60]. Theoretically, $\beta$ equals $d/6$ with $d$ the effective dimension of the domain growth. We obtained a $\beta$ value of 0.29-0.32 for WZ'-$In_2Se_3$, which aligns well with 0.33 predicted for a 2D domain growth ($d = 2$). Because the FE polarization switching is driven by domain growth via the movement of domain walls, a 2D domain growth suggests the notable presence of IP domain walls, with polarization reversal occurring through the sequential switching of each quintuple layer. Such IP domain walls have been visualized recently in ZB' type $\alpha$-$In_2Se_3$ crystals[18], their behavior in WZ' type $\alpha$-$In_2Se_3$ film warrants further study in the future.

Although a direct correlation between the nonlinear current switching behavior and the IP FE polarization switching has been revealed, we mention that these current peaks are not solely from the displacement current during FE polarization switching, but are also associated with the accompanying charge injection and subsequent trapping at the Pt/$In_2Se_3$ interface. On one hand, this is supported by the fact that the switching charge density $Q$ ($Q = \frac{1}{A}\int_0^T Idt = \frac{1}{A}\frac{dt}{dV}\int_0^V IdV$ with $A$ the cross-section area of the device) is as large as 5 mC/$cm^2$ (inset of Fig. 4a & Supplementary Fig. S16), far exceeding the theoretical IP polarization of ~115 μC/$cm^2$ for WZ'-$In_2Se_3$[12]. On the other hand, it is supported by the potential change upon the IP voltage writing as investigated by Scanning Kelvin Probe Force Microscopy (SKPFM, Fig. 4d). In the SKPFM amplitude image, the Pt electrode with a high work function will display a relatively low potential compared to that of $In_2Se_3$. However, there are abnormal regions on the electrode as indicated by the blue dashed box, where the potential is increased by ~0.05−0.1 V (see also Fig. S17). This can be attributed to charge trapping in $In_2Se_3$, which alternates the electric properties of the $In_2Se_3$/Pt interface. This is reasonable considering that $In_2Se_3$ is an n-type semiconductor with energy levels of donors being ~0.05−0.09 eV below the conduction band minimum[61]. We thus conclude that the current switching characteristics arose largely from two aspects: the ferroelectric effect, in which FE surface bounding charges asymmetrically modulated the Schottky barriers (Fig. 4e), and charge trapping effect, in which electric-field induced charge injection and trapping modulated the Schottky barrier (Fig. 4f). The asymmetric modulation was like the ferroelectric switchable diode effect[6], where positive

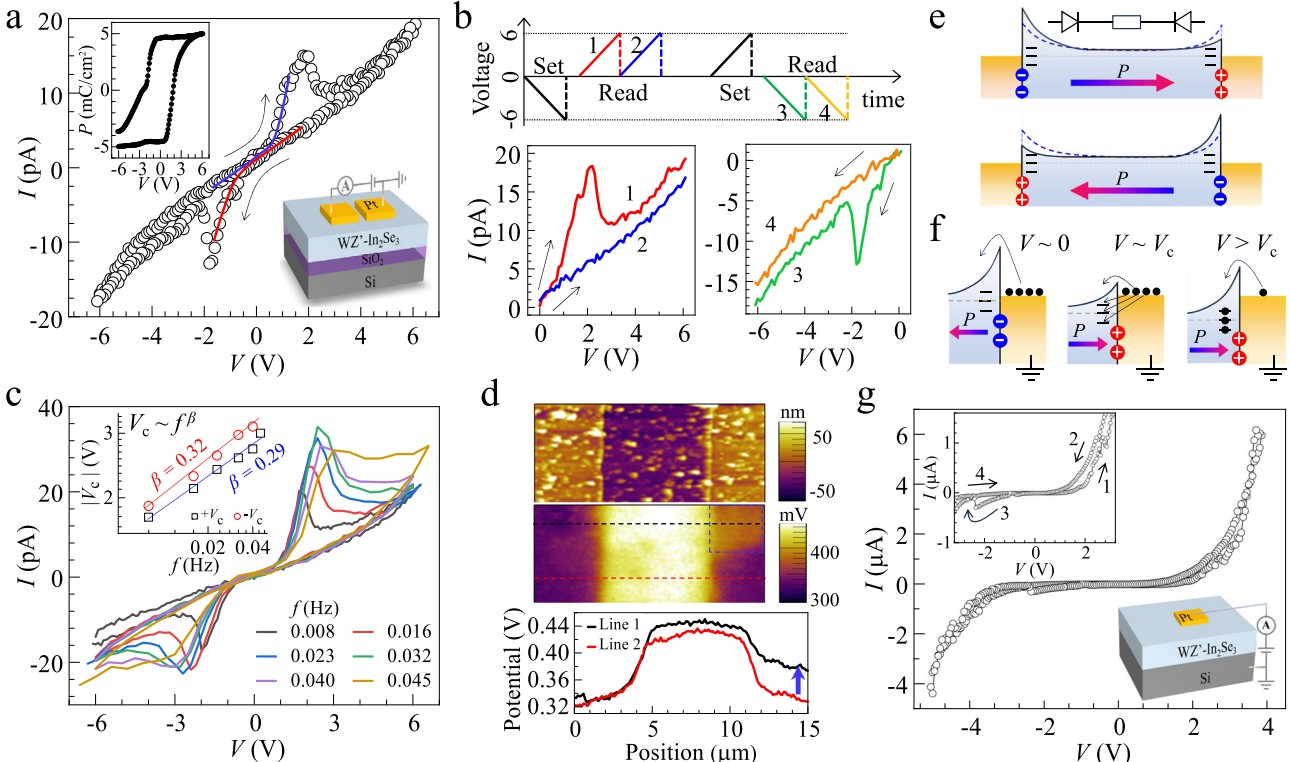

**Fig. 4 | I–V characteristics of two terminal devices based on WZ' type α-In$_2$Se$_3$ films. a** I–V curve of a Pt/In$_2$Se$_3$/Pt in-plane device with a channel width of 6 μm. Insets show the nominal polarization loop (up) derived from the I–V curve, and the schematical device structure (down). Black arrows indicate the voltage sweeping direction. Blue and red lines are guide to the eyes. **b** Consecutive I–V sweepings after setting the device by a −6 V and a +6 V triangle voltage pulse. Arrows indicate the voltage scan direction. **c** I–V curves with various scanning rates. Inset represents the peak voltage $V_c$ as a function of the sweeping frequency, which follows a traditional power low as indicated by red and blue lines. **d** SKPFM surface potential image of a typical Pt/In$_2$Se$_3$/Pt IP device and the line scans along the black and red dashed lines. **e** Schematic diagrams for energy bands with polarization pointing to opposite directions. The device can be regarded as two back-to-back diodes in series with a bulk resistance, and FE polarization can alter the Schottky barriers. **f** The band evolutions of right In$_2$Se$_3$/Pt interface under various positive voltages applied onto the left electrode. **g** I–V curve of the out-of-plane (OOP) Pt/In$_2$Se$_3$/Si device. The top inset is the enlarged I–V curve with numbered arrows indicating the voltage sweeping sequence, and the bottom inset is a schematic of the OOP device.

(negative) polarization charge can reduce (enhance) the band bending of FE semiconductors because of charge screening effect. This gives rise to the reverse diode behavior upon polarization reversal as indicated by red and blue lines in Fig. 4a. To explain the current peaks, let us focus on the right In$_2$Se$_3$/Pt interface with polarization pointing to the left. When a small positive voltage is applied onto the left Pt electrode, the current transport is mainly controlled by the right In$_2$Se$_3$/Pt interface which is reverse-biased, and the current flow is minor because of the up-bending barrier induced by negative polarization charge (Fig. 4f). With voltage approaching +$V_c$, FE polarization flips and the positive polarization charge reduces the barrier, and hence the electron can inject into In$_2$Se$_3$ more easily and the corresponding current increases significantly. In the meantime, partially injected electrons will fill the donor traps and compensate for the positive polarization charge, resuming the barrier gradually and making the subsequent electron injection more difficult. Therefore, the current decreases with further increasing the voltage, resulting in the current peak near +$V_c$. The peak on the negative voltage bias can also be explained by a similar mechanism. Impressively, the current switching phenomenon remains evident up to 180 days postfabrication of the device (Supplementary Fig. S18), indicating a considerable FE stability of WZ' type α-In$_2$Se$_3$ film in ambient conditions at room temperature. Similar current peaks, though relatively subtle due to the predominant leakage current, were also observed in the OOP device (Fig. 4g), suggesting the robust OOP FE polarization within the film. We also fabricated an IP Pt/ZB'-In$_2$Se$_3$/Pt device using a ZB' type α-In$_2$Se$_3$ film deposited on a mica substrate. The I–V curves exhibit minor

hysteresis (Supplementary Fig. S19), with no observable current switching peaks—a stark contrast to the behavior of WZ'-In$_2$Se$_3$-based device (Fig. 4a). This difference likely arises from the weaker IP ferroelectric polarization of ZB'-In$_2$Se$_3$. As previously discussed, stronger ferroelectric polarization facilitates more pronounced charge trapping/de-trapping during polarization reversal, enhancing the current switching behavior.

The conductance switching in IP devices suggests the ability of achieving continuous conductive states via voltage pulsing, a feature particularly appealing for integrating IP devices, such as artificial neural synapses, into brain-inspired computing architecture. In such devices, the synaptic connection strength can be modified efficiently by using voltage as the external stimulus, hence emulating various fundamental synaptic plasticity including long-term potentiation (LTP) and long-term depression (LTD), which are crucial for realizing functions of learning and memory[32]. As shown in Fig. 5a, typical LTP and LTD synaptic behaviors were realized electronically by applying positive and negative voltage pulse sequences onto the IP device, both in the dark and under light illumination (532 nm with an intensity of 58 mW/cm$^2$). Thanks to the large light absorption coefficient, the light illumination can significantly enhance conductance ($G$) by 10 times and promote the dynamic $G$ ratio from 0.5 to 2. Importantly, both LTP and LTD are stable in the cycling test, which gives a cycle-to-cycle variation of 6.1% under light illumination (Fig. 5b), better than that of ~8% in the dark (inset of Fig. 5b).

Empirically, LTP and LTD can be described by an exponential function with a nonlinear factor $v$, which is a key parameter that

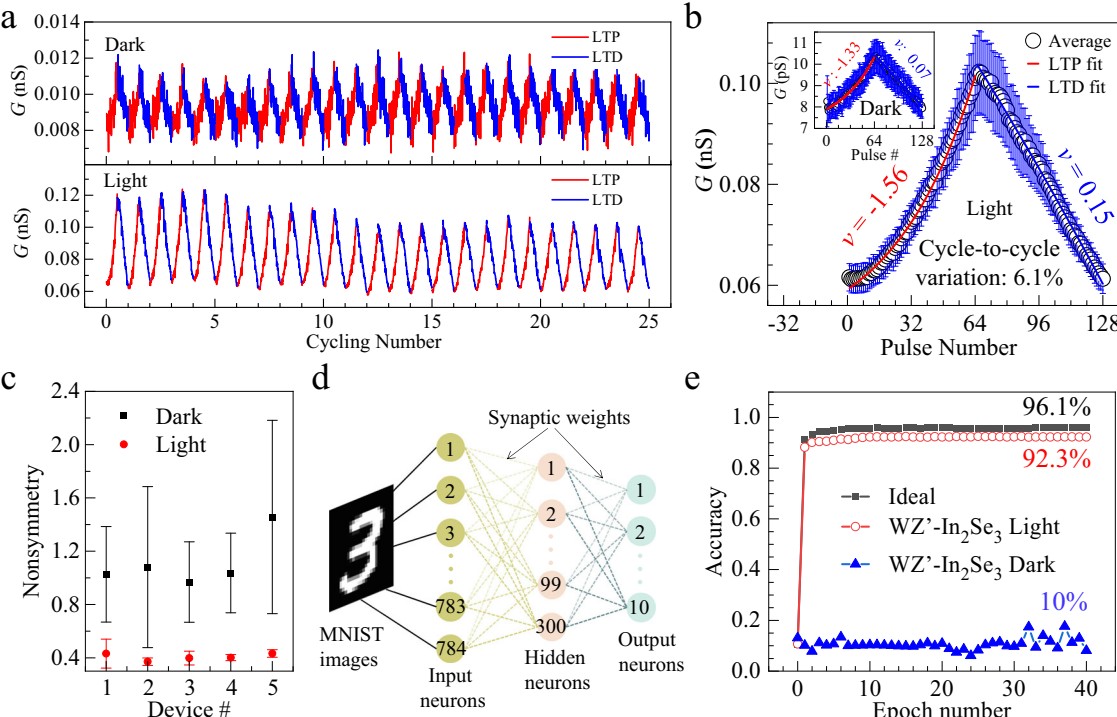

**Fig. 5 | In-plane synapse device for supervised learning. a** 25 cycles of repetitive LTP and LTD operations using an incremental pulse scheme. **b** The averaged conductance changes versus pulse number. Error bars indicate the standard deviation from which the cycle-to-cycle variations can be determined. Solid lines are the exponential fittings by using the nonlinear factor $v$. **c** The non-symmetry factors of 5 randomly selected devices. For each device, 5 cycles of LTP and LTD are averaged, and the error bars represent the standard deviation. **d** The designed neural network for implementing the backpropagation algorithm. **e** The training accuracies of WZ' type $\alpha$-In$_2$Se$_3$-based ANNs for the MNIST images. Ideal device-based ANN is for comparison.

ensures the consistent relationship between the activity of post-neurons and pre-neurons. From the linearity fittings (solid lines in Fig. 5b), we obtained a $v$ of −1.56 and 0.15, respectively, for LTP and LTD processes under light illumination, which is comparable with that of −1.33 and 0.07 for LTP and LTD in the dark. In addition, the trajectory of the weight-increasing process in LTP usually differs from that of the weight-decreasing process in LTD, a characteristic known as non-symmetry (NS), which significantly influences the plasticity of learning and memory functions. We performed LTP and LTD characterizations for 5 cycles on 5 randomly selected devices, from which we derived the NS values of ~0.4 for devices operated under light illumination, better than that of ~1.0 operated in dark (Fig. 5c). The good device metrics of $G$ ratio, linearity, and symmetry promoted by light illumination, allows us to perform a supervised pattern classification task using a simulated artificial neural network (ANN) hardware based with these WZ' type $\alpha$-In$_2$Se$_3$-based synapses. Specifically, a multilayer perceptron (MLP) type ANN (Fig. 5d) was built to execute the backpropagation algorithm based on the conductance updates in Fig. 5a. The 28 × 28-pixel image version of handwritten digits from the "Modified National Institute of Standards and Technology (MNIST)" data set was employed for the pattern classification task. As shown in Fig. 5e, the classification accuracy of these digit images exceeds 90% on the second epoch and achieves an accuracy as high as 92.3% after 40 training epochs, close to the ideal floating point-based neural network performance of 96.1%. In contrast, the accuracy is only ~10 % for devices operated in the dark, because of their small dynamic $G$ ratio and large cycle-to-cycle variation, suggesting the importance of photon illumination in enhancing the classification accuracy.

## Discussion

Although the WZ' type $\alpha$-In$_2$Se$_3$ has been theoretically predicted, its experimental realization has remained elusive. Given that the WZ' and

ZB' phases have comparable formation energies, the emergence of the WZ' phase in our case indicates that kinetic factors−rather than purely thermodynamic considerations−play a dominant role in our fabrication process. Conventionally, ZB' type $\alpha$-In$_2$Se$_3$ is synthesized via CVD, physical vapor deposition (PVD), or chemical vapor transport (CVT), where crystalline In$_2$O$_3$ and Se powders serve as precursors. In contrast, our approach employs amorphous In$_2$O$_3$ precursor film deposited at room temperature by PLD (Fig. 1d). The subsequent reaction between amorphous In$_2$O$_3$ and Se powder differs from that involving the crystalline In$_2$O$_3$ and Se powder, leading to the formation of the WZ' phase. This is further corroborated by a control experiment using crystalline In$_2$O$_3$ film deposited by PLD at 750$^\circ$ as the precursor, which yields $\gamma$-In$_2$Se$_3$ instead, as evidenced by its characteristic Raman peak at 150 cm$^{-1}$ (Supplementary Fig. S20). A notable advantage of our method lies in PLD's ability to produce ultrasmooth amorphous In$_2$O$_3$ films. This enables the direct synthesis of continuous In$_2$Se$_3$ film on substrates such as SiO$_2$ and Si, in stark contrast to the nanoflakes produced by CVD alone, as depicted in Fig. S1. While continues In$_2$Se$_3$ films have previously been limited to mica substrates, our approach significantly enhances the compatibility of In$_2$Se$_3$ with silicon-based electronics, offering a practical route for device integration.

While ferroelectricity in WZ'-type In$_2$Se$_3$ has been demonstrated by PFM, TEM, and electric transport measurements, quantifying IP ferroelectric polarization in 2D ferroelectrics remains challenging. This difficulty stems primarily from their semiconducting nature, which often involves high carrier densities−a particular issue for narrow-bandgap materials like WZ'- In$_2$Se$_3$ ($E_g$ ~ 0.8 eV). Our transient $I−V$ measurements using a ferroelectric tester on Pt/WZ'-In$_2$Se$_3$/Pt IP device (Supplementary Fig. S21) revealed current switching kinks or peaks similar to those in Fig. 4, with $V_c$ following a power-law frequency dependence ($V_c \propto f^\beta$, $\beta = 0.25(3)$), indicative of 2D domain growth. However, the overwhelmed non-ferroelectric leakage current

results in the open-mouthed *P-V* loops with unrealistically large polarization values. This demonstrates that quantification of IP polarization in 2D ferroelectrics requires alternative measurement techniques.

In summary, we have successfully developed an in-situ transport growth method by combining PLD and CVD to fabricate the centimeter-scale, continuous WZ' type $\alpha$-In$_2$Se$_3$ films directly on silicon or silicon dioxide substrates. This marks the inaugural experimental preparation of WZ' type $\alpha$-In$_2$Se$_3$, confirmed as a 2D FE semiconductor with a high $T_c$ exceeding 620 K and a tunable bandgap of down to 0.8 eV. The bandgap exhibits a strong reduction from 1.6 eV to 0.8 eV as the film thickness increases from 3 nm to 25 nm, primarily attributed to the charged domain walls within the film. Thanks to the small bandgap, WZ' type $\alpha$-In$_2$Se$_3$ exhibits excellent light absorption ability compared to conventional 3D semiconductors such as Si and GaAs. By taking advantage of FE polarization switching, we show that the two-terminal IP devices based on WZ' type $\alpha$-In$_2$Se$_3$ exhibit memristive switching that can mimic the synaptic functions artificially. The good light absorption promotes the dynamic *G* range, linearity, and symmetry of the synapse, leading to a high recognition accuracy of 92.3% in a supervised pattern classification task. Our finding adds a new phase to the 2D FE semiconductor family and demonstrate its potential in memory device applications.

## Methods

### Growth of 2D WZ' type $\alpha$-In$_2$Se$_3$

Amorphous In$_2$O$_3$ thin films were deposited on $1 \times 1$ cm$^2$ SiO$_2$ (5 μm)/Si (or Si) substrates at room temperature and in vacuum ($1 \times 10^{-5}$ Pa) by pulsed laser deposition (PLD) using a KrF excimer laser ($\lambda = 248$ nm). A laser energy density of 1.5 J/cm$^2$ at a repetition rate of 1 Hz was used. In$_2$O$_3$ film and a boat of Se block were then placed into a two-zone CVD furnace. In$_2$O$_3$ was placed in the downstream and about 15 cm away from the boat of Se. The temperatures of Se and In$_2$O$_3$ film were heated to 270 °C and 610 °C, respectively, in 40 min and were kept for 30 min to grow 2D WZ' type $\alpha$-In$_2$Se$_3$ under a pressure of 1 atm in the atmosphere of 10 sccm Ar and H$_2$ mixture. After the growth, the substrate was cooled down to 350 °C at a rate of 4 °C/min and kept there for 30 min to remove excess Se adhering to the surface. Finally, the furnace was naturally cooled down to room temperature within 30 min.

### Structure characterizations

The crystal structure of the film was characterized by X-ray diffraction (Bruker D8). The EDS data were collected by an Oxford energy dispersive spectrometer matched with a tungsten hairpin filament scanning electron microscope (Quanta 600). The cross-sectional TEM samples were fabricated using the focused ion beam (FIB) (Thermal Fisher Helios G4). The atomic structures were investigated at 200 kV using a probe-corrected scanning transmission electron microscopy (Thermofisher Spectra 300) equipped with Gatan Continuum 1065. During the experiment, the collection angle of the detector was set to be ~50−200 mrad, and the convergence angle was set to be ~21.6 mrad.

### Optical characterizations

The optical images and surface of In$_2$Se$_3$ films were taken by a Nikon LV-ND100 optical microscope and a Bruker Dimension Icon AFM system, respectively. The Raman measurement was performed using an excitation laser of 532 nm (LabRAM HR Evolution, HORIBA Jobin Yvon) equipped with a low-wavenumber filter. The spectral resolution is 0.65 cm$^{-1}$. The transmission spectra of WZ'-In$_2$Se$_3$ thin film with a thickness of 18 nm grown on a fused silica substrate were carried out using a UV−visible spectrometer (U-3010) with a spectral resolution of 0.1 nm. Photoluminescence (PL) measurement was performed using a fluorescence spectrophotometer (Hitachi F-4500) with an excitation wavelength of 320 nm and a spectral resolution of 0.2 nm.

### Raman and first-principles calculations

First-order Raman intensity as a function of phonon frequency $\omega_\mu$ and laser excitation energy $E_L$ is calculated by the third-order time-dependent perturbation theory as follows,

$$I^\mu\left(\omega_\mu, E_L\right) \propto \left| \sum_{\mathbf{k}} \sum_{i=f,m,m'} \frac{\boldsymbol{M}_{\mathbf{opt}}^{fm'}(\mathbf{k}) \cdot \boldsymbol{M}_{\mathbf{ep},\mu}^{m'm}(\mathbf{k}) \cdot \boldsymbol{M}_{\mathbf{opt}}^{mi}(\mathbf{k})}{\left(E_L - \Delta E_{mi} - \mathrm{i}\gamma\right)\left(E_L - \hbar\omega_\mu - \Delta E_{fm'} - \mathrm{i}\gamma\right)} \right|^2$$

(1)

where $\boldsymbol{M}_{\mathbf{opt}}$ and $\boldsymbol{M}_{\mathbf{ep}}$ are electron-photon coupling and electron-phonon coupling matrix elements, respectively. To simulate Raman spectra, the electronic band structures and the phonon dispersion of In$_2$Se$_3$ were calculated by generalized-gradient approximation (GGA) function with the electronic exchange-correlation functional of the Perdew−Burke−Ernzerhof (PBE) type as implemented in Quantum ESPRESSO package[62]. Relativistic norm-conserving pseudopotentials derived from an atomic Dirac-like equation and a 120 Ry kinetic energy cutoff for the plane-wave basis were used. The atomic structures were fully relaxed until the atomic force is less than $10^{-5}$ Ry/Bohr. The $12 \times 12 \times 4$ **k**-grid and $6 \times 6 \times 2$ **q**-grid Monkhorst-Pack mesh were used to sample the Brillouin zone (BZ) during the calculation of electronic band structures and the phonon dispersion of In$_2$Se$_3$. Then, based on the Wannier interpolation schemes as implemented in the EPW code[63], the electron-photon and electron-phonon coupling matrix elements with only Γ-point phonon involved were calculated on the $36 \times 36 \times 12$ **k**-grid in the whole BZ. Finally, the Raman spectra were calculated and analyzed using the home-made QR$^2$-code[34].

### Electronic band structure calculations

First-principles density functional calculations were performed using plane wave basis sets with the projector-augmented wave (PAW) method, as implemented in the VASP code[64]. The exchange and correlation functional were treated using the Perdew−Burke−Ernzerhof (PBE) parameterization of generalized gradient approximation (GGA)[65]. The plane-wave energy cut-off was set at 350 eV, and structural relaxations were conducted until the interatomic forces were below 0.01 eV/Å, while electronic step self-consistency was ensured with energy convergence better than $10^{-5}$ eV. Given the critical influence of van der Waals (vdW) interactions, interlayer vdW forces were appropriately included using the widely applied DFT-D3 empirical correction as implemented in VASP[66]. In this vdW correction scheme, the atom-pairwise specific dispersion coefficients and cutoff radii are both computed from first principles. The Herd−Scuseria−Ernzerhof hybrid functional (HSE06) was employed to calculate the electronic band structures of WZ'-type $\alpha$-In$_2$Se$_3$ and related structures with domains using the k-point meshes $16 \times 16 \times N$, with $N = 8, 4, 2$ and 1 for bulk WZ'-In$_2$Se$_3$, bulk WZ'-In$_2$Se$_3$ with NDW, WZ'-In$_2$Se$_3$ with CDW and monolayer, respectively.

### Ferroelectric characterization

SHG testing was performed by a homebuilt inverted microscope with a broadband Ti: sapphire oscillator (Vitara, Coherent, Inc.) delivering 15 fs laser pulses centered at 1040 nm and a repetition rate of 80 MHz. The SHG spectrum was collected through a spectrometer (Horiba JY 1250 M) with a resolution of 0.06 nm. The piezo-response force microscopy (PFM) analysis of the films was performed using a commercial atomic force microscopy (Asylum Research MFP-3D) with Pt/Ir-coated Si cantilever tips (SCM-PIT). In typical PFM measurements, an ac driving voltage of 1 V at ~300 kHz was applied to the tip. The local piezoelectric hysteresis loops were carried out using a DART (dual a.c. resonance tracking) mode with the external DC writing voltages sweeping between −12 V and +12 V.

## Device fabrication and electrical characterizations

For electric measurement, FE diodes were fabricated by photolithography followed by sputtering the Pt electrodes with a thickness of 50 nm and a size of $50 \times 50$ μm$^2$ on In$_2$Se$_3$ films. The I-V curves were recorded using a multi-source meter (Keithley 2450) equipped with a probe station (Lake Shore, EMPX-H2). For in-plane tests, two probes were placed onto the two neighboring Pt electrodes with a gap of 6 μm. For out-of-plane tests, Pt was the top electrode and the heavily doped silicon substrate acted as the bottom electrode. LTP and LTD characterizations in the dark were performed by measuring the conductance at 2 V bias after sequential voltage pulses increased from 3 V to 5 V for the LTP process and -3 V to -5 V for the LTD process, respectively. The pulse width was fixed at 1 ms. For electric tests under light illumination, a 532 nm laser with a light intensity of 58 mW/cm$^2$ was used.

## LTP and LTD linearity and symmetry analysis

To quantitatively analyze the LTP and LTD performance, we use an empirical model to fit the $G$ change ($G_p$ for potentiation and $G_d$ for depression processes) with the number of pulses ($N$) through the following equations[67],

$$G_p = G_{min} + G_0(1 - e^{-vn}) \tag{2}$$

$$G_d = G_{max} - G_0[1 - e^{-v(n-1)}] \tag{3}$$

$$G_0 = (G_{max} - G_{min})/(1 - e^{-v}) \tag{4}$$

where $G_{max}$ and $G_{min}$ are the maximum and minimum conductance directly extracted from the experimental data. $n$ is the normalized pulse number $n = N/N_{max}$ with $N_{max}$ the maximum pulse number used in the LTP or LTD processes. $v$ is the nonlinearity factor that refers to the linearity of the curve relating the device conductance to the number of programming pulses, which can be positive or negative depending on the convexity of the curve. Good linearity (with $v < 1$) ensures a straightforward and consistent relationship between the activity of post-neurons and pre-neurons, which can simplify neural network simulations, minimize noise interference, and boost the stability and efficiency of neural networks, thereby enhancing the precision of learning and memory processes.

In addition to the linearity, the trajectory of the weight increase process in LTP usually differs from that of the weight decrease process in LTD, referring as the asymmetry. It will also limit the recognition accuracy, which was defined as:

$$NS = \frac{\max|G_P(N) - G_d(129 - N)|}{G_P(128) - G_p(1)} \text{ for } N = 1 \text{ to } 64 \tag{5}$$

Where $G_P(N)$ and $G_d(N)$ are the conductance values after the $N^{th}$ potentiation pulse and $N^{th}$ depression pulse, respectively. $NS = 0$ corresponds to a completely symmetric weight update.

## Supervised learning

The neural network simulations were carried out using a CrossSim simulator based on the backpropagation algorithm[68]. In this simulator, a multilayer perceptron (MLP) composed of 784 input neurons, 300 hidden neurons, and 10 output neurons was used. Each synaptic weight matrix between two neuron layers was implemented by a memristor crossbar. The synaptic weight was mapped onto the conductance of the memristor. Each memristor possesses a tunable conductance that can be modulated by applying voltage pulses, following the LTP and LTD curves. When performing the inference, an input image was first converted to a vector of voltage pulses whose amplitudes were proportional to the image pixel values. Then, these voltage pulses were fed to the input neurons and subsequently applied to the rows of the memristor crossbar. The output currents along the columns represented the result of the matrix-vector multiplication. During the training, the weights were tuned following the experimentally measured LTP/LTD characteristics, under the guidance of the backpropagation algorithm. We constructed a $784 \times 300 \times 10$ network for recognizing images with $28 \times 28$ pixels from the "Modified National Institute of Standards and Technology" (MNIST) dataset. The MNIST dataset contains 60000 and 10000 images for the training and test, respectively. The learning rate for the training on the MNIST dataset was optimized to be 0.2.

## Data availability

The authors declare that all the data supporting the results of this study can be found in the paper and its Supplementary Information file. The detailed data for the study is available from the corresponding authors upon request.

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

## Acknowledgements

This work is supported by the National Key R&D Program of China (Grant No. 2022YFA1203903) (W.J.H.), the National Natural Science Foundation of China (NSFC) (Grant Nos. 61974147, 52031014, 92477120, 52402133) (W.J.H., Z.D.Z., W.J.H., Y.Z.L.), International Partnership Program of

Chinese Academy of Sciences (CAS) (Grant No. 172GJHZ2024044MI) (W.J.H.), the Strategic Priority Research Program of CAS (Grant No. XDB0460000) (T.Y.), and Special Fund for Central Government Guiding the Local Development of Science and Technology (Grant No. 2023JH6/100100063) (W.J.H.). Thanks to Prince Sultan University for computational support for band structure calculations.

## Author contributions

Y.X.J. and W.J.H. conceived and designed the experiment. W.J.H. and Z.D.Z. supervised the project. Y.X.J. conducted the thin film growth, device fabrication, and measured their optical and electric transport properties with the help of Y.Z.L., B.H.H., and B.L. Y.X.J. prepared the TEM sample, X.K.N. and K.P. Song carried out the HRSTEM experiments. R.H.L. conducted the Raman spectra calculation, S. Ali performed electronic band structure calculations under the supervision of TY. C.B.Q. and Q.K.L. conducted the SHG measurements. Y.X.J., B.H.H., and J.H.Q. performed the PFM measurement and analysis with help from X.F.Z. and G. Liu. Q.K.L. carried out the SKPFM measurement. H.Y.D. and Z.F. performed the artificial neural network simulation. Y.X.J. and W.J.H. wrote the manuscript, F.X., T. Yang, G. Liu, and L.J.L. helped improve the manuscript. Y.X.J., X.K.N., R.H.L., and K.P.S. contributed equally to this work.

## Competing interests

The authors declare no competing interests.

## Additional information

[1]Shenyang National Laboratory for Materials Science, Institute of Metal Research, Chinese Academy of Sciences, Shenyang, China. [2]School of Materials Science and Engineering, University of Science and Technology of China, Shenyang, China. [3]Hebei Key Lab of Optic-Electronic Information and Materials, National and Local Joint Engineering Research Center of Metrology Instrument and System, Hebei University, Baoding, China. [4]Electron Microscopy Center, School of Chemistry & Chemistry Engineering, Shandong University, Jinan, China. [5]Energy, Water, and Environment Lab, College of Humanities and Sciences, Prince Sultan University, Riyadh, Saudi Arabia. [6]Institute for Advanced Materials and Guangdong Provincial Key Laboratory of Optical Information Materials and Technology, South China Academy of Advanced Optoelectronics, South China Normal University, Guangzhou, China. [7]School of Physical Science and Technology, Jiangsu Key Laboratory of Frontier Material Physics and Devices, Soochow University, Suzhou, China. [8]State Key Laboratory of Quantum Optics Technologies and Devices, Institute of Laser Spectroscopy, Shanxi University, Taiyuan, China. [9]Collaborative Innovation Center of Extreme Optics, Shanxi University, Taiyuan, China. [10]Center for Quantum Matter, School of Physics, Zhejiang University, Hangzhou, China. [11]Department of Materials Science and Engineering, National University of Singapore, Singapore, Singapore. [12]These authors contributed equally: Yuxuan Jiang, Xingkun Ning, Renhui Liu, Kepeng Song. ✉e-mail: yangteng@imr.ac.cn; wjhu@imr.ac.cn

