## [Transparent Peer Review file · Nature Communications]

2D ferroelectric narrow-bandgap semiconductor Wurtzite-type α -In₂Se₃ and its silicon-compatible growth

Corresponding Author: Professor Weijin Hu

Version 1:

Reviewer comments:

Reviewer #1

(Remarks to the Author)

This manuscript is dedicated to the discovery of a novel two-dimensional In₂Se₃ variant, characterized by a distorted Wurtzite (WZ') crystal structure. Although this structure has been previously predicted theoretically, it has not yet been experimentally verified. The authors present abundant experiment data to elucidate the properties of this new phase, including its high-resolution crystal structure, its ferroelectricity with a notable Curie temperature of 620 K, a narrow band gap down to 0.8 eV, and impressive light absorption that comparable to those of leading semiconductor materials. The findings are truly exciting, particularly in light of the extensive research already conducted on α -In₂Se₃. Overall, I would rate this work as a remarkable contribution to the field of 2D ferroelectrics, which is bound to stimulate a wealth of research interest in this new phase. I will recommend its publication without any reservation, pending a few minor revisions to refine the presentation and enhance the clarity.

-In the introduction section, I would recommend the authors to incorporate the content of sliding ferroelectricity, to provide a comprehensive reflection of the recent development in the field of 2D ferroelectrics.

-It would be beneficial for the authors to incorporate additional explanations on their success in fabrication WZ' variant. Given that prior research has largely been centered on the ZB' structure. This will enhance the unique aspects of their approach and highlight the significance of their findings in the context of the current literature.

-PLD is commonly employed for the deposition of oxide materials. The combination of PLD with CVD to fabricate 2D materials is notable. It would be informative to know if attempts were made to produce the variant solely with CVD, which could provide further insight into the relative merits of each technique in this context.

-I will suggest the authors to change the word "in-situ growth" into "in-situ transport growth", to maintain consistency with the corresponding term "remote transport growth." This change would more effectively capture the essence of the CVD process, as vapor transport is inherently involved.

-As revealed by theoretical calculation, this novel phase has a big in-plane polarization of 115 $\mu\text{C}/\text{cm}^2$. I am quite curious whether they have conducted measurements of this quantity using conventional ferroelectric testing methods, such as those typically applied to ferroelectric perovskites with a traditional ferroelectric tester?

-Have the authors attempted to determine the atomic positions of the In and Se atoms from TEM? It would be optimal if they could calculate the polarization of this novel variant based on the precise atomic positions.

-The recent experimental work (Ref. 18) has revealed the domain structure of ZB'-In₂Se₃. Have the authors investigated the domain structure of WZ' variant to provide a comparative analysis?

- The PFM amplitude depicted in Fig. 2h is indeed impressive, with the domain wall characterized by an absence of amplitude response clearly visible. To my knowledge, such an observation is quite rare in PFM studies of 2D ferroelectrics. However, with respect to the quantitative determination of the piezoelectric coefficient d_{33} as shown in Fig. S8, the measurements conducted near the resonant peak may potentially introduce artifacts. I will suggest the authors perform additional measurements to verify the reliability of the reported d_{33} values.

-Typo errors, list one for example. "This discrepancy might arise from the spin-orbit coupling (SOC) of In atoms, whose strength can alter the position of valence band maximum (VBM) within the k-space, potentially causing a transition from a direct to an indirect bandgap²⁹." Here "whose" should be "whose". I would suggest the authors to check any additional typo errors and grammar mistakes throughout the manuscript.

Reviewer #2

(Remarks to the Author)

In the article, the authors report the growth and characterisation of the wurtzite (WZ) phase of α -In₂Se₃. They fabricate and characterise devices based on the grown material. The growth of this new phase would be interesting due to its predicted enhanced ferroelectric properties compared to the more common zinc-blend (ZB) structure. The successful growth and demonstration of devices based on WZ- α -In₂Se₃ would thus be interesting to the community looking to create ferroelectric devices. However, in its current form, the article cannot be considered suitable for publication. In particular, I have several concerns on the conclusions made about the grown material, and I would like more details and discussion to be provided on these questions before the article could be further.

1) In Figure 1j the Raman spectra for the sample is shown, and in the text it is compared to the known Raman spectrum of ZB α -In₂Se₃. The peak positions, as noted by the authors, are in very similar positions to that of the more conventional ZB phase and the reported differences are very slight. In order that these can be compared easily, the authors should report in the text the quantitative differences between each peak position rather than needing to check with the figure and the table in the supplementary material. The authors cite reference 12 where there is comparison of phonon dispersion but not Raman spectra. The phonon spectra in Ref 12 of ZB and WZ look extremely similar, and it is difficult to determine the expected differences in the peak positions. The authors should provide more details of the comparison between the theoretically obtained peak positions (of all the peaks) to their measured peak positions. Ideally, the authors could provide the theoretical Raman spectrum of two phases of the crystal to compare to their data.

2) Due to the broadening of the peaks and the difference in the peak positions observed at different parts of the sample, the authors should provide an estimate of the uncertainty in the Raman peak. The broadening of the peaks appears to be quite large, presumably due to disorder, leading to the possibility that for the lowest peak they could be observing a combination of the E₂ and A₁ peaks in a disordered sample. The authors should also comment on the significance of the differences in the peak position measured at different places on the sample.

3) A much larger difference between the reported spectra and those expected of the ZB phase is the peak at $\sim 150 \text{ cm}^{-1}$, which is much larger than typically observed for ZB α -In₂Se₃. However, a large peak at $\sim 150 \text{ cm}^{-1}$ is expected for the γ phase, see ref 20 of the manuscript. Might this peak indicate there are mixed regions of each of these phases?

4) The authors report an indirect band gap of 0.8 eV and show that it compares well to the calculated band gap given in Ref. 29 of the text. However, the 0.8 eV gap in Ref 29, was calculated using the GGA-PBE approximation which underestimates the band gap. The calculations in Ref. 12 show more accurate calculations of both phases using HSE06, which gives a band gap $\sim 1.5 \text{ eV}$ for both ZB and WZ, a value which compares well with measurements on ZB phase, suggesting that it would be difficult to differentiate between the phases via the band gap measurement. The calculations in Ref. 12 therefore do not support the measured value of 0.8 eV being associated with the WZ phase. Also, the authors report a direct gap of $\sim 3 \text{ eV}$, this is much larger than that reported in the calculations, which would not be easily explained by the inclusion of spin orbit interaction. The authors should comment further on the comparison of their data with these calculations and explain this discrepancy with Ref. 12.

5) The authors should provide a modified version of figure 2b, of the HRSTEM data. They should clearly show how the quintuple layers would stack with an interlayer vdW gap to create the bulk heterostructure. It appears in the schematic of the suggested atom positions that the upper-most atoms of the quintuple layer have been placed so that there is not a significant gap between the quintuple layers. This is in contrast to the structures presented in this article:

<https://pubs.acs.org/doi/10.1021/acs.inorgchem.8b01950>, where there is a significant vdW space gap between the layers.

6) The authors describe the ferroelectric properties of the grown material. They appear to attribute most of the strong ferroelectric effect to trapped charges rather than the intrinsic properties of the material itself. Given that the intrinsic ferroelectric effect is not as important, the authors should expand on whether, for this type of mechanism, the WZ phase is a more effective material than the ZB for these types of devices.

Version 2:

Reviewer comments:

Reviewer #1

(Remarks to the Author)

The authors have well addressed my concerns and revised the manuscript accordingly. I would recommend the manuscript for publication.

Reviewer #2

(Remarks to the Author)

I am satisfied that the authors have answered my initial questions on their manuscript very thoroughly. They have undertaken substantial new work and detailed further analysis which makes their new conclusions more convincing. The new HRTEM figures with supporting DFT calculations are illuminating and, along with the band-gap calculation, support the interesting proposal of the emergence of charged domain walls. I recommend that this article can be published after the authors have considered the additional points below.

- I would suggest that the authors state the quantitative difference in formation energies of the different phases discussed on line 291.

- For full clarity, the authors should state the nature of the vdW correction scheme used in the calculations.

- I believe that the authors should be careful about statements made about the band gap of the Wurtzite type being 0.8 eV. Their own DFT calculations suggest that the bulk Wurtzite type in its simplest form has a band gap of 1.37 eV. It is only with the consideration of the formation of domain walls (and a hybrid structure is used within the DFT calculations) when a band gap of 0.8 eV is obtained. Are the authors proposing that the Wurtzite phase will always form with the proposed domain structure, or is this a result of the growth conditions of this work? Perhaps a sentence such as "the band gap can be tuned to"... or similar would be appropriate? I would also be careful about the statement in the abstract about this point, where for clarity the domain walls should be mentioned.

Version 3:

Reviewer comments:

Reviewer #2

(Remarks to the Author)

I am satisfied that the authors have answered all of my questions and suggestions and I recommend this article for publication.

Response to reviewers' comments

Dear Reviewers,

Thank you very much for your thorough review and critical comments on our manuscript “2D ferroelectric narrow-bandgap semiconductor Wurtzite’ type α - In_2Se_3 and its silicon-compatible growth” (NCOMMS-25-05276A-Z). Below we provide a point-to-point response to address your concerns. We note that your comments are in *blue italic*, words in black font are our responses, and changes made to the manuscript are in red. We believe your comments have helped us improve the manuscript significantly.

Reviewer #1:

This manuscript is dedicated to the discovery of a novel two-dimensional In_2Se_3 variant, characterized by a distorted Wurtzite (WZ’) crystal structure. Although this structure has been previously predicted theoretically, it has not yet been experimentally verified. The authors present abundant experiment data to elucidate the properties of this new phase, including its high-resolution crystal structure, its ferroelectricity with a notable Curie temperature of 620 K, a narrow band gap down to 0.8 eV, and impressive light absorption that comparable to those of leading semiconductor materials. The findings are truly exciting, particularly in light of the extensive research already conducted on α - In_2Se_3 . Overall, I would rate this work as a remarkable contribution to the field of 2D ferroelectrics, which is bound to stimulate a wealth of research interest in this new phase. I will recommend its publication without any reservation, pending a few minor revisions to refine the presentation and enhance the clarity.

Our Reply: Thank you for your thoughtful and encouraging feedback. Your recognition means a lot to us. We will address your comments diligently to improve the manuscript.

1. In the introduction section, I would recommend the authors to incorporate the content of sliding ferroelectricity, to provide a comprehensive reflection of the recent development in the field of 2D ferroelectrics.

Our Reply: This is a very good point. Sliding ferroelectricity is an emerging phenomenon that is unique to bilayer or multilayer two-dimensional (2D) materials, as well as their heterostructures. It arises from the interlayer charge redistribution and transfer between neighboring layers within these 2D materials. The concept of sliding ferroelectricity was initially proposed by *L. Li et al.* in their groundbreaking study [*ACS Nano*, **11**, 6382, 2017]. Since then, sliding ferroelectricity has been experimentally verified in various 2D materials, such as few-layered WTe_2 , bilayer BN, and β - InSe . In recognition of its significance, we have incorporated this topic into the introduction of our revised manuscript as follows: “**Various fascinating phenomena have emerged at the 2D level, ...the sliding/twisting ferroelectrics**

stemming from charge redistribution during the interlayer translation or twisting between bilayer or multilayer 2D materials, ...". And accordingly, we have cited the following references:

[1] Li, L. & Wu, M. Binary compound bilayer and multilayer with vertical polarizations: two-dimensional ferroelectrics, multiferroics, and nanogenerators. *ACS Nano*, **11**, 6382 (2017).

[2] Zheng, Z. *et al.* Unconventional ferroelectricity in moiré heterostructures. *Nature* **588**, 71-76 (2020).

[3] Yasuda, K., Wang, X., Watanabe, K., Taniguchi, T. & Jarillo-Herrero, P. Stacking-engineered ferroelectricity in bilayer boron nitride. *Science* **372**, 1458-1462 (2021).

[4] Sui, F. *et al.* Sliding ferroelectricity in van der Waals layered γ -InSe semiconductor. *Nat. Commun.* **14**, 36 (2023).

2. It would be beneficial for the authors to incorporate additional explanations on their success in fabrication WZ' variant. Given that prior research has largely been centered on the ZB' structure. This will enhance the unique aspects of their approach and highlight the significance of their findings in the context of the current literature.

Our Reply: Thank you for raising this important question. We are pleased to report the first successful experimental synthesis of the WZ' phase. However, there remains substantial work ahead to fully understand its properties. Considering that WZ' phase has the similar formation energy to its ZB' counterpart, the preferred formation of WZ' phase in our case suggests that in our specific case, kinetic factors played a more predominant role than thermodynamic ones in our fabrication process. In contrast to previous studies, which have produced the ZB' type α -In₂Se₃ via chemical vapor deposition (CVD), physical vapor deposition (PVD), or chemical vapor transport (CVT) methods, our approach employed amorphous In₂O₃ thin films that were deposited at room temperature using pulsed laser deposition (PLD), as depicted in **Figs. R1a & R1c**. This stands in contrast to the crystalline starting materials (In₂O₃ and Se powder) used in conventional methods. The subsequent reaction between amorphous In₂O₃ and Se powder in our process may differ entirely from that involving crystalline In₂O₃ and Se powder, potentially leading to the formation of WZ' phase. To test this hypothesis, we conducted a control experiment by using a crystalline In₂O₃ film fabricated by PLD at a growth temperature of 750 °C, as shown in **Fig. R1b**. Subsequent Raman spectroscopy analyses (**Fig. R1d**) confirmed the formation of γ -In₂Se₃ following selenization. **We have included these additional findings into the discussion section of the revised manuscript.**

Fig. R1. In-situ transport growth of In_2Se_3 by combining PLD and CVD. (a) The XRD of amorphous In_2O_3 prepared by PLD and (c) the Raman spectra of resulting WZ'-type $\alpha\text{-In}_2\text{Se}_3$ after selenization. (b) The XRD of crystalline In_2O_3 prepared by PLD and (d) the Raman spectra of resulting $\gamma\text{-In}_2\text{Se}_3$ after selenization.

3. *PLD is commonly employed for the deposition of oxide materials. The combination of PLD with CVD to fabricate 2D materials is notable. It would be informative to know if attempts were made to produce the variant solely with CVD, which could provide further insight into the relative merits of each technique in this context.*

Our Reply: Thank you for your recognition of our work. We have dedicated considerable time to the study of oxide ferroelectrics using PLD, and we have also previously ventured into the realm of CVD to synthesize ZB'-type $\alpha\text{-In}_2\text{Se}_3$. This interdisciplinary foundation has enabled us to integrate PLD and CVD techniques to investigate 2D materials, as presented in this manuscript. To our knowledge, few studies have adopted such an integrative approach. We have attempted to fabricate WZ' phase exclusively using CVD, yet the outcomes indicate that under identical CVD conditions, we obtained the $\beta\text{-In}_2\text{Se}_3$ or $\gamma\text{-In}_2\text{Se}_3$ phases, as shown in **Fig. R2**, which align with the findings of previous research. The reason behind, as we respond to your comment #2, could be attributed to the amorphous In_2O_3 film we used instead of crystalline In_2O_3 powder commonly employed. A notable advantage of our approach is the ability of PLD to fabricate the smooth amorphous In_2O_3 film very conveniently. This precursor film can react

with Se power directly on the substrate, allowing for the direct formation of the continuous In_2Se_3 film on substrates such as SiO_2 and Si. This is in stark contrast to the rougher films produced by CVD alone, as depicted in **Fig. R2**. Given that continuous In_2Se_3 films were previously only synthesized on mica substrates, our approach offers a significant benefit for the integration of In_2Se_3 into silicon-based electronic devices. **We have added these data and analysis into the discussion section in the revised manuscript.**

Fig. R2. In_2Se_3 deposited on amorphous SiO_2 substrate by CVD solely. The optical image (a) and Raman spectra (b) of $\beta\text{-In}_2\text{Se}_3$, and the optical image (c) and Raman spectra (d) of $\gamma\text{-In}_2\text{Se}_3$ phase.

4. *I will suggest the authors to change the word “in-situ growth” into “in-situ transport growth”, to maintain consistency with the corresponding term “remote transport growth.” This change would more effectively capture the essence of the CVD process, as vapor transport is inherently involved.*

Our Reply: Thank you for your careful reading and good suggestion. **We have changed it into “in-situ transport growth (ITG)” in the revised manuscript.**

5. *As revealed by theoretical calculation, this novel phase has a big in-plane polarization of 115 $\mu\text{C}/\text{cm}^2$. I am quite curious whether they have conducted measurements of this quantity using conventional ferroelectric testing methods, such as those typically applied to ferroelectric*

perovskites with a traditional ferroelectric tester?

Our Reply: Yes, we have performed transient I - V and the corresponding P - V hysteresis measurements using a ferroelectric tester (Radiant Multiferroic) on an in-plane Pt/In₂Se₃/Pt device, which featured a gap width of 4 μm and a length of 50 μm (**Fig. R3**). We applied a range of voltages across different frequencies, from 10 Hz down to 0.1 Hz. The observed asymmetric transient current in response to the voltage bias indicates that the two interfaces between Pt and In₂Se₃ are different, despite the use of identical Pt electrodes. This asymmetry is likely due to the intrinsic built-in electric field resulting from the in-plane ferroelectric polarization. The switching of polarization is evident from the kinks appearing in the transient I - V curves, as marked by the red arrows. However, due to the significant leakage currents at high frequencies, the corresponding P - V loops are open-mouthed, and the polarization values appear unrealistically high, complicating the determination of the intrinsic in-plane ferroelectric polarization values through electrical measurements. This challenge is a common issue encountered with 2D ferroelectric semiconductors, particularly with WZ'-type In₂Se₃, which has a narrow bandgap as low as 0.8 eV. Nevertheless, the ferroelectricity can still be inferred from the frequency-dependent coercivity (V_c). As represented in **Fig. R3h**, V_c determined from the kinks in the I - V curves follows a power law with frequency f ($V_c \sim f^\beta$), where the exponent β is estimated to be $\sim 0.25(3)$. This value is in agreement with the range of 0.29-0.32 reported for the device discussed in the main text (**Fig. 4c**).

Fig. R3. I - V curves and the corresponding P - E loops measured by ferroelectric tester. a-c. Transient current loops at 10 Hz, 1 Hz, and 0.1 Hz measured by a ferroelectric tester. d. Quasistatic I - V curve measured by Source Meter for comparison. e-g. The corresponding P - E loops at 10 Hz, 1 Hz, and 0.1 Hz. h. The coercive voltage (V_c) at different frequencies. V_c is determined from the kink of I - V curves. Red solid line is the fitting according to the power law $V_c \sim f^\beta$ with β equaling to 0.25(3).

Action: We have included an additional paragraph into the discussion section of the revised manuscript to address this challenge as follows: “While ferroelectricity in WZ’-type In_2Se_3 has been demonstrated by PFM, TEM, and electric transport measurements, quantifying IP ferroelectric polarization in 2D ferroelectrics remains challenging. This difficulty stems primarily from their semiconducting nature, which often involves high carrier densities—a particular issue for narrow-bandgap materials like WZ’- In_2Se_3 ($E_g \sim 0.8$ eV). Our transient I-V measurements using a ferroelectric tester on Pt/WZ’- In_2Se_3 /Pt IP device (Supplementary Fig. S21) revealed current switching kinks or peaks similar to those in Fig. 4, with V_c following a power-law frequency dependence ($V_c \propto f^\beta$, $\beta = 0.25(3)$), indicative of 2D domain growth. However, the overwhelmed non-ferroelectric leakage current results in the open-mouthed P - V loops with unrealistically large polarization values. This demonstrates that quantification of IP polarization in 2D ferroelectrics requires new measurement techniques. ”

6. Have the authors attempted to determine the atomic positions of the In and Se atoms from TEM? It would be optimal if they could calculate the polarization of this novel variant based on the precise atomic positions.

Our Reply: Yes. From the HRTEM images, we determined the relative positions of In and Se atoms, as illustrated in Fig. R4a, where the distances represent statistical values averaged over 10 atom columns in the HRTEM image. For comparison, Fig. R4b shows the theoretically predicted structure obtained from first-principles calculations using the HSE06 method. The experimental and theoretical atomic positions exhibit good agreement within experimental error. This will give polarization values of $\sim 1.8 \mu\text{C}/\text{cm}^2$ in the OOP direction and $\sim 115 \mu\text{C}/\text{cm}^2$ in the IP direction, as have been done in the seminal work by W. J. Ding *et al.* (*Nat. Commun.* **8**, 14956, 2017). We don’t perform additional theoretical calculations on these values.

Figure R4. The atomic positions of In and Se atoms for WZ’ type α - In_2Se_3 . a. determined from HRTEM image. b. determined from first-principles calculations.

7. The recent experimental work (Ref. 18) has revealed the domain structure of ZB'-In₂Se₃. Have the authors investigated the domain structure of WZ' variant to provide a comparative analysis?

Our Reply: Thank you for your question. **Fig. R5** presents the cross-sectional HRTEM image of a typical atomic resolved domain configurations of WZ' phase. As indicated by the numbers, there are five stacking layers with varied atomic configurations. Layers 1, 3, and 5 exhibit the characteristic atomic configuration of the WZ' phase, each with opposite OOP polarizations denoted by black arrows. These ferroelectric domains are separated by in-plane domain walls (IP-DWs) of varying types: layer 2 represents a head-to-head IP-DW, while layer 4 is a tail-to-tail IP-DW. These IP-DWs display a uniform non-polar state, with the central Se atoms positioned in the center of the cell. This configuration results in a sharp flip of the polarization vector direction, with the IP-DW width measuring a single unit cell along the c-axis (~ 0.8 nm) — even narrower than that observed in conventional three-dimensional ferroelectrics like BiFeO₃ (~ 2 nm; *Nat. Mater.* **8**, 229, 2009). Furthermore, we conducted a statistical analysis of the off-center displacements of the central Se atoms within the domain. The displacement, measured as the distance from the central Se atoms to their symmetric position along the c-axis, is ~ 0.3 - 0.4 Å as illustrated in **Fig. R5b**. This value aligns well with the theoretical value of ~ 0.4 Å. **We have included Fig. R5 and the analysis into the revised manuscript.**

Fig. R5. Atomically resolved characterization of the ferroelectric domain configuration in WZ'-type α -In₂Se₃. **a.** Five typical layers with van der Waals gaps of ~ 1 - 1.4 Å are clearly seen. Each layer comprises five atoms arranged in an atomic sequence of Se-In-Se-In-Se, which are bounded by covalent bonds. WZ' phase of different polarization directions (layer 1, 3, and 5) are isolated by In-plane domain walls (IP DW, layer 2 and 4). Their atomic microstructures are schematically shown in typical regions marked by rectangles of different colors. **b.** The displacement of the central Se atom in each layer along the c axis. 10-unit cells have been counted in each layer. Inset schematically shows the IP-DWs and out-of-plane polarizations.

8. The PFM amplitude depicted in Fig. 2h is indeed impressive, with the domain wall characterized by an absence of amplitude response clearly visible. To my knowledge, such an observation is quite rare in PFM studies of 2D ferroelectrics. However, with respect to the quantitative determination of the piezoelectric coefficient d_{33} as shown in Fig. S8, the measurements conducted near the resonant peak may potentially introduce artifacts. I will suggest the authors perform additional measurements to verify the reliability of the reported d_{33} values.

Our Reply: Thank you for your valuable suggestion. PFM is the widely accepted method for quantifying the piezoelectric properties of ferroelectrics at the nanoscale, particularly for 2D ferroelectrics. According to the reviewer's advice, we have carried out d_{33} measurements at six additional, randomly chosen positions across the film. The results are presented in **Fig. R6**. Resonant piezoelectric peaks have been observed in all the cases, with their amplitudes increasing with the driving ac voltage (V_{ac}) applied onto the probe and can be well fitted by using the harmonic oscillator model (solid lines). The derived intrinsic piezoelectric amplitudes as a function of V_{ac} are shown in **Fig. R6h**, where the PFM amplitudes linearly increase with V_{ac} , yielding d_{33} values that span from 1.8 pm/V to 4.6 pm/V. These findings are in agreement with our earlier measurement, which reported a d_{33} value of 2 pm/V. **We have included these additional data into the revised manuscript to enhance the reliability of reported d_{33} values.**

Fig. R6. Piezoelectric coefficient d_{33} of WZ' type α -In₂Se₃ film. a-g. The piezoelectric response near the contact resonant frequency under different driving voltages on seven randomly selected positions in the film. Solid lines are fittings according to the harmonic oscillator model. **h.** The piezoelectric amplitude in unit of pm as a function of the driving voltage derived from **a-g**. **i.** d_{33} values derived from the slope of **h**.

9. Typo errors, list one for example. "This discrepancy might arise from the spin-orbit coupling (SOC) of In atoms, whole strength can alter the position of valence band maximum (VBM) within the k-space, potentially causing a transition from a direct to an indirect bandgap²⁹." Here "whole" should be "whose". I would suggest the authors to check any additional typo errors and grammar mistakes throughout the manuscript.

Our Reply: Thank you for pointing out this grammar mistake. We have thoroughly reviewed the manuscript and have corrected all possible grammar mistakes. We appreciate your insightful feedback and comments, and we believe these additional data and clarifications have substantially enhanced the quality of the manuscript.

Reviewer#2:

In the article, the authors report the growth and characterization of the wurtzite (WZ) phase of α -In₂Se₃. They fabricate and characterize devices based on the grown material. The growth of this new phase would be interesting due to its predicted enhanced ferroelectric properties compared to the more common zinc-blend (ZB) structure. The successful growth and demonstration of devices based on WZ- α -In₂Se₃ would thus be interesting to the community looking to create ferroelectric devices. However, in its current form, the article cannot be considered suitable for publication. In particular, I have several concerns on the conclusions made about the grown material, and I would like more details and discussion to be provided on these questions before the article could be further:

Our Reply: Thank you for your interest and for recognizing the value of our work. Below we will try our best to address your specific concerns related to the conclusions made about the material, and provide more discussions accordingly. With the additional data and analysis provided below, I sincerely hope that you will recommend it for publication.

1. In Figure 1j the Raman spectra for the sample is shown, and in the text it is compared to the known Raman spectrum of ZB α -In₂Se₃. The peak positions, as noted by the authors, are in very similar positions to that of the more conventional ZB phase and the reported differences are very slight. In order that these can be compared easily, the authors should report in the text the quantitative differences between each peak position rather than needing to check with the figure and the table in the supplementary material. The authors cite reference 12 where there is

comparison of phonon dispersion but not Raman spectra. The phonon spectra in Ref 12 of ZB and WZ look extremely similar, and it is difficult to determine the expected differences in the peak positions. The authors should provide more details of the comparison between the theoretically obtained peak positions (of all the peaks) to their measured peak positions. Ideally, the authors could provide the theoretical Raman spectrum of two phases of the crystal to compare to their data.

Our Reply: Thank you for raising this important question. Raman spectra, which are highly sensitive to atomic vibrations and crystal structures, including lattice constants, bonding lengths, and bonding angles, have been extensively utilized for the phase determination of 2D materials. Despite this, the close structural similarity among the various phases of In_2Se_3 poses a challenge for discrimination using Raman spectra. According to your suggestion, we have conducted theoretical Raman calculations on the two phases of $\alpha\text{-In}_2\text{Se}_3$. Due to the minute differences between these phases, an accurate calculation of Raman spectra necessitates the consideration of double-resonant Raman scattering process, beyond the conventional single-resonant Raman process. We have developed this approach and the accompanying QR²-code code in our previous research [1], and successfully used it to identify typical 2D materials including graphene and transition metal dichalcogenides (TMDs) [2-6]. As depicted in the upper panel of **Fig. R7**, the calculated Raman spectra for both the WZ' and ZB' types exhibit remarkable similarity, as anticipated. However, a distinct redshift is observable in the Raman peaks of the WZ' type relative to that of the ZB' type. Specifically, the wavenumber of the E^2 mode decreases from $\sim 88 \text{ cm}^{-1}$ for the ZB' phase to $\sim 85 \text{ cm}^{-1}$ for the WZ' phase from our calculation. Moreover, the primary peak of WZ' can be effectively resolved into two distinct peaks corresponding to the E^2 mode and A_1^1 mode, whereas there is an overlap in the position of the two peaks in the ZB' phase. These subtle differences are corroborated by the experimental Raman spectra of the WZ' phase (middle panel of **Fig. R7**) and the ZB' phase (lower panel of **Fig. R7**). The experimental blue shift is up to $\sim 11 \text{ cm}^{-1}$, which is roughly four times greater than the theoretical prediction, facilitating the experimental differentiation of the two types.

Although the theoretical calculations successfully captured the principal red-shift phenomenon between the two phases, there are some discrepancies between the calculations and experiments. First, the specific Raman peak positions are different between them. Second, in contrast to the WZ' phase, the Raman peaks at $\sim 150 \text{ cm}^{-1}$ and $\sim 242 \text{ cm}^{-1}$ are nearly absent in the ZB' phase. These discrepancies may be due to the thermal excitations that can dampen certain vibration modes at higher temperatures, considering that the theoretical calculations were performed at 0 K. Additionally, interlayer couplings in thicker In_2Se_3 films may also contribute to these differences.

Fig. R7. Raman spectra of WZ'-type and ZB'-type In_2Se_3 . Top panel, theoretical calculation. Middle panel, experimental spectra of WZ'- In_2Se_3 . Bottom panel, experimental spectra of ZB'- In_2Se_3 .

[1] Huang, J. *et al.* QR²-code: An open-source program for double resonance Raman spectra. arxiv:2505.10041 (2025). <https://doi.org/10.48550/arXiv.2505.10041>. and QR²-code, <<http://qr2-code.com/>> (2024).

[2] Huang, J. *et al.* First-principles calculations of double resonance Raman spectra for monolayer MoTe₂. *Phys. Rev. B* **105** (2022). <https://doi.org/10.1103/PhysRevB.105.235401>

[3] Liu, R. *et al.* Helicity selection rule of double resonance Raman spectra for monolayer MoSe₂. *Phys. Rev. B* **110** (2024). <https://doi.org/10.1103/PhysRevB.110.245422>

[4] Pang, Y., Huang, J., Yang, T. & Zhang, Z. Accurate assignment of double resonant Raman bands in Janus MoSSe monolayer from first-principles calculations. *Jour. Mater. Sci. & Tech.* **131**, 82-90 (2022). <https://doi.org/https://doi.org/10.1016/j.jmst.2022.05.022>

[5] Zhang, S. *et al.* Quantum interference directed chiral Raman scattering in two-dimensional enantiomers. *Nat. Commun.* **13**, 1254 (2022). <https://doi.org/10.1038/s41467-022-28877-6>

[6] Zhang, Y. *et al.* DUV Double-Resonant Raman Spectra and Interference Effect in Graphene: First-Principles Calculations. *J. Raman Spectrosc.* **56**, 316-323 (2025). <https://doi.org/10.1002/jrs.6768>

Action: We have included the theoretical Raman spectra calculations, and added the following sentences accordingly into the revised manuscript “Raman spectroscopy reveals information on the vibration modes of atoms which is very sensitive to the crystal structures including lattice constant, bonding length and angles, and so has been extensively utilized for the phase determination of 2D materials such as In_2Se_3 ^{20,33,34}. Nevertheless, due to their similar crystal structures, and sometimes low quality of the samples, controversial results are often reported in

literatures. We thus conducted theoretical Raman calculations on these phases by using our custom developed QR²-code³⁵ (details see **Methods**), which has been successfully used to identify typical 2D materials including graphene and transition metal dichalcogenides (TMDs)³⁶⁻⁴⁰. We considered not only the conventional single-resonant Raman process but also the double resonant scattering process to get a more accurate result as represented in the upper panel of **Fig. 1j**. Both WZ' and ZB' phases exhibit remarkable similarity as anticipated. However, a distinct frequency red-shift is observable in the Raman peaks of the WZ' phase relative to the ZB' phase. Specifically, the wavenumber of the E² mode decreases from ~88 cm⁻¹ for the ZB' phase to ~85 cm⁻¹ for the WZ' phase. Moreover, the primary peak of the WZ' phase can be effectively resolved into two distinct peaks corresponding to the A₁¹ mode and E² mode, whereas these modes are overlapping in the ZB' phase. These subtle differences are corroborated by the experimental Raman spectra of the WZ' phase (**Fig. 1j**, middle panel) and ZB' phase on mica (**Fig. 1j**, lower panel). The experimental red-shift is up to ~11 cm⁻¹, which is roughly four times greater than the theoretical prediction, facilitating the experimental differentiation of the two phases. Additionally, the intensity ratio of E³/(E² + A₁¹) is ~0.71 in experiment, which aligns closely with the theoretical value of ~0.6 (as shown in **Fig. S7**). This agreement reinforces the dominant formation of WZ' phase in our case. To conclude, the theoretical calculations have captured the dominant red-shift feature of Raman spectra from the ZB' phase to the WZ' phase, however, there are some discrepancies between the calculations and experiments. First, the theoretical Raman peak positions are generally positioned at a smaller wavenumber compared to that of experimental spectra. This is due to the choice of pseudopotentials, specifically generated by the projector augmented wave (PAW) method with the generalized gradient approximation (GGA) for exchange-correlation functional, which typically induces under-binding interatomic potentials and consequently reduces vibrational frequencies. Second, the Raman peaks at ~149 cm⁻¹ and ~241 cm⁻¹ are nearly absent in the experimental spectra of ZB' phase. These discrepancies may be due to the thermal excitations that can dampen certain vibration modes at higher temperatures, considering that the theoretical calculations were performed at 0 K. Additionally, interlayer couplings in thicker In₂Se₃ films could also contribute to these differences. Further investigation is required to elucidate the detailed mechanism underlying these observations.”

2. Due to the broadening of the peaks and the difference in the peak positions observed at different parts of the sample, the authors should provide an estimate of the uncertainty in the Raman peak. The broadening of the peaks appears to be quite large, presumably due to disorder, leading to the possibility that for the lowest peak they could be observing a combination of the

E2 and A1 peaks in a disordered sample. The authors should also comment on the significance of the differences in the peak position measured at different places on the sample.

Our Reply: Thank you for your suggestion. In response to your previous comment, we have demonstrated that the lowest E^2 and A_1^1 vibration peaks are indeed theoretically overlapping. The broadening of these peaks further contributes to their overlap. Nevertheless, by using the professional software Labspec, we are able to deconvolute the Raman peaks due to the E and A modes, and determine the position of each peak and its peak shape characterized by Full width at half maximum (FWHM). We have performed Raman spectra measurement on nine randomly selected positions across the film. The raw Raman spectra are represented in **Fig. 2j**, while the baseline-subtracted Raman spectra are shown in **Fig. R8a**. Peak fitting was carried out for each spectrum, with a representative example illustrated at the bottom, displaying the fitted peaks. We have summarized the statistics of peak positions and their FWHMs as error bars in **Fig. R8b**. A minor discrepancy between the peaks is observed, indicating the film's homogeneity. It is worth noting that the Raman peak at $\sim 200 \text{ cm}^{-1}$ is attributed to the overlapped Raman peaks of the E^4 , A_1^2 , and A_1^3 modes, as predicted by theoretical calculations. Consequently, it is more challenging to ascertain definitive peak positions compared to other peaks.

Fig. R8. Raman spectra statistics. **a.** the Raman spectra acquired on 9 positions randomly selected in the film with the baseline being subtracted. The fitted peaks with Lorentz functions are shown at the bottom for a typical Raman spectrum at position #1. **b.** The statistics of the Raman peaks. Due to the overlapping, the peaks of E^4 , A_1^2 , and A_1^3 show more distinct fluctuations compared with other Raman peaks.

Action: We have added **Fig. R8** and related discussion into the revised manuscript.

3. A much larger difference between the reported spectra and those expected of the ZB phase is the peak at $\sim 150 \text{ cm}^{-1}$, which is much larger than typically observed for ZB $\alpha\text{-In}_2\text{Se}_3$. However, a large peak at $\sim 150 \text{ cm}^{-1}$ is expected for the γ phase, see ref 20 of the manuscript. Might this peak indicate there are mixed regions of each of these phases?

Our Reply: Thank you for raising this interesting question. As our response to your comment #1, the theoretically calculated Raman spectra show a distinct Raman peak at $\sim 150 \text{ cm}^{-1}$ both for WZ'-type and ZB'-type $\alpha\text{-In}_2\text{Se}_3$. This Raman peak relates to the E^3 mode which represents two degenerate in-plane vibration modes of the top Se-In along and perpendicular to the edge of the hexagonal lattice. The relative peak intensity of E^3 over that of E^2 and A_1^1 is ~ 0.71 experimentally, which is highly consistent with the theoretical value of ~ 0.60 , suggesting it is dominated by WZ' phase rather than γ phase (**Fig. R9**). Additionally, we didn't see any clue of γ phase in STEM characterizations. Third, $\gamma \text{In}_2\text{Se}_3$ is a direct bandgap semiconductor with a bandgap of $\sim 1.95 \text{ eV}$ for bulk [2D Mater. 5, 035026, 2018], which is much larger than that of $\sim 0.8 \text{ eV}$ in our case. We thus conclude that the Raman peak at 150 cm^{-1} is dominated by WZ' type $\alpha\text{-In}_2\text{Se}_3$ rather than $\gamma\text{-In}_2\text{Se}_3$ in our case.

Fig. R9. The intensity ratio of E^3 over that of E^2 and A_1^1 derived from experimental Raman spectra (solid circles) and theoretical spectra (blue dashed line).

4. The authors report an indirect band gap of 0.8 eV and show that it compares well to the calculated band gap given in Ref. 29 of the text. However, the 0.8 eV gap in Ref 29, was calculated using the GGA-PBE approximation which underestimates the band gap. The calculations in Ref. 12 show more accurate calculations of both phases using HSE06, which

gives a band gap ~ 1.5 eV for both ZB and WZ, a value which compares well with measurements on ZB phase, suggesting that it would be difficult to differentiate between the phases via the band gap measurement. The calculations in Ref. 12 therefore do not support the measured value of 0.8 eV being associated with the WZ phase. Also, the authors report a direct gap of ~ 3 eV, this is much larger than that reported in the calculations, which would not be easily explained by the inclusion of spin orbit interaction. The authors should comment further on the comparison of their data with these calculations and explain this discrepancy with Ref. 12.

Our Reply: Thank you for your critical comments and suggestions. After looking through the seminal work referenced in Ref. 12, we acknowledge the variation in bandgap calculations for In_2Se_3 , which differ depending on the theoretical methods employed, such as GGA-PBE and HSE06. Ref. 29 has also explored the impact of spin-orbit coupling (SOC) on the bandgap characteristics. It is noteworthy that both studies focused on single-layer In_2Se_3 with a thickness of ~ 0.8 nm, whereas our experimental study involves multilayer In_2Se_3 with a thickness of 18 nm. To clarify your question on the bandgap values, we have fabricated WZ'- In_2Se_3 films with varying thickness ranging from 3 nm to 25 nm on quartz substrates, and systematically investigated their transmission spectra, as illustrated in **Fig. R10a**. The corresponding absorption coefficients, depicted in **Fig. R10b**, exhibit a gradual reduction with increasing film thickness, yet maintain a high value of over 10^6 cm^{-1} at around 300 nm. Using Tauc plots $(\alpha h\nu)^{1/2}$ (left) and $(\alpha h\nu)^2$ (right) vs. $h\nu$, we have determined the indirect bandgap and direct bandgap for films of different thickness, as shown in **Fig. R10h** and **Fig. R10i**, respectively. We found that the indirect bandgap of 3 nm In_2Se_3 is ~ 1.6 eV, which is in close agreement with the ~ 1.5 eV predicted for single layer α - In_2Se_3 using the HSE06 method [*Nat. Commun.* **8**, 14956, 2017]. However, the bandgap reduces rapidly to ~ 0.8 eV as the film thickness exceeds 18 nm, demonstrating a pronounced thickness dependence. This behaviour is consistent with observations in various 2D material systems, including transition metal dichalcogenides (TMDCs) like MoS_2 and PtSe_2 , etc, which are often attributed to the quantum confinement, interlayer couplings, and dielectric screening effect. [*Physica E.* **109**, 11, 2019]. Note that while a direct bandgap of ~ 3 eV was inferred from the Tauc plot of $(\alpha h\nu)^2$ vs. $h\nu$ in our experiments, the Tauc plot is an empirical tool that should be complemented with other characterization methods to ascertain the bandgap types. For direct bandgap semiconductors, the direct recombination of electrons and holes results in efficient photon emission, typically manifesting as a distinct PL peak near the bandgap edge. However, the absence of PL peaks in our films (**Fig. R11**) suggests that WZ'-type In_2Se_3 is an indirect bandgap semiconductor.

Fig. R10. Thickness dependent optical properties of WZ'-type α -In₂Se₃. a. the transmission spectrum within the wavelength range of 200 nm to 800 nm. b. the as derived optical absorption coefficient as a function of wavelength. c-g. $(\alpha h\nu)^{1/2}$ (left) and $(\alpha h\nu)^2$ (right) Tauc plots to determine the band gaps for films with thickness ranging from 3 nm to 25 nm. Solid lines are the fittings. h. The as-derived indirect bandgaps, and g. the direct bandgaps.

Fig. R11. PL spectrum of WZ'-type In₂Se₃ with thickness ranging from 3 nm to 25 nm. An excitation photon energy of 3.88 eV has been used. The peak located at 1.94 eV is due to the

grating interference of the light source, which has also been observed for the SiO₂/Si substrate. No PL peak was detected within the photon energy range of 1.4 eV to 3.4 eV, indicating that the direct E_g of ~ 3.1 eV derived from the empirical Tauc plot is not favored for WZ'- α -In₂Se₃.

To clarify the difference between the WZ' and ZB' type α -In₂Se₃, we have summarized their thickness-dependent bandgaps in **Fig. R12**. Although both phases are theoretically predicted to be indirect bandgap semiconductors, ZB' type α -In₂Se₃ is considered to have a direct bandgap due to its distinct PL peak, [*Adv. Funct. Mater.* **30**, 2004206, 2020] from which the bandgaps are determined. In stark contrast to the WZ' phase, the bandgap of ZB' type α -In₂Se₃ exhibits a relatively slow increase from ~ 1.405 eV to ~ 1.457 eV as the thickness decreases from 800 nm to 10 nm, this trend is consistent with the theoretical prediction of ~ 1.47 eV.

Fig. R12. Thickness dependent bandgaps of α -In₂Se₃. The experimental data of ZB' phase (solid diamond) is determined from PL spectra reported in Ref. [*Adv. Funct. Mater.* **30**, 2004206, 2020], the theoretical data of single-layer In₂Se₃ is from Ref. [*Nat. Commun.* **8**, 14956, 2017]. The shaded lines are guide to the eyes.

To unravel the origin of thickness-dependent bandgaps, we performed electronic band calculations by using HSE06. Three cases have been considered: (i) 1L, 3L, and bulk WZ'-In₂Se₃ without including the effect of domain wall (**Fig. R13a**), (ii) bulk WZ'-In₂Se₃ with neutral domain wall (NDW, **Fig. R13b**), and (iii) bulk WZ'-In₂Se₃ with charged domain wall (CDW, **Fig. R13c**). We found that when domain wall is not considered, the bandgap of WZ' phase reduces slightly from ~ 1.62 eV for 1L to ~ 1.37 eV for bulk. While if one considers the NDW, the bandgap still keeps at a high value of ~ 1.45 eV. Whereas, CDW can effectively reduce the bandgap down to ~ 0.96 eV (**Fig. R13c**), in line with the experimental value of ~ 0.8 eV. These results strongly suggest that a correct description of this distinct thickness dependence should consider the effect of CDW, which has been microscopically revealed by

HRTEM in Fig. 2b. Furthermore, we conducted projected band structure calculations for bulk WZ' type α -In₂Se₃ by including NCW (Fig. R14a) and CDW (Fig. R14b). Compared with NCW, where the conduction band minimum is dominated by the s orbitals of In of the WZ' layer (Fig. R14a); we found that s orbitals of In atoms in CDW can contribute an additional conduction band located closer to the Fermi level, thus reducing the bandgap efficiently. This fact suggests that the charges accumulated in the CDW due to the charge screening effect, tend to make the domain wall more conducting. Similar effect has been observed for conventional ferroelectric oxides like BiFeO₃, where the CDW can reduce the bandgap by 0.25-0.5 eV, which is usually attributed to the combined effect of defects like oxygen vacancies and the charge screening effect inside the domain wall [Phys. Rev. Lett. 105, 197603, 2010].

Fig. R13. Calculated electronic band structures of WZ' type α -In₂Se₃. (a) 1L, 3L, and bulk without considering the domain wall. (b) after considering the neutral domain wall (NDW). NDW is sandwiched between two neighbouring WZ' layers with the same polarization

direction. (c) after considering the charged domain wall (CDW). CDW is sandwiched between two neighbouring WZ' layers with opposite polarization directions.

Fig. R14. Calculated projected electronic band structures of bulk WZ' type α - In_2Se_3 . (a) with neutral domain wall, (b) with charged domain wall.

Finally, we note that CDW is typically considered unstable due to its high depolarization field and associated electrostatic energy. In conventional ferroelectrics like BiFeO_3 , CDW can be stabilized via the charge screening, structure reconstruction, or domain wall movement. In our case, the total energy of the WZ' phase with CDW (~ -3.632 eV/atom) is slightly higher than that of NDW (~ -3.634 eV/atom), suggesting instability. However, we find that electrostatic interactions between neighbouring CDWs with opposite charges can reduce the energy by ~ 4 meV/atom, thereby stabilizing the CDW state (**Fig. R15**). Specifically, this attractive Coulomb energy gain from NDW to CDW can be estimated by $Q^2/(4\pi\epsilon_r)$, where CDW net charge $Q = \rho \times (4/3)\pi d^3$ with $\rho \sim 8.6 \times 10^{-5}$ e/Bohr³, $d \sim 2.66$ Å (bond length of In-Se), the inter-CDW distance $r \sim 15$ Å. While this long-range inter-CDW energy gain may go beyond the Coulomb Ewald sum, it remains significant in terms of the energy precision of 10^{-5} eV in our calculation.

Fig. R15. Calculated interlayer charge re-distribution $\Delta\rho$ of bulk WZ' type α -In₂Se₃ by considering (a) the neutral domain walls (NDWs), and (b) the charged domain walls (CDWs). $\Delta\rho = \rho(\text{WZ}' + \text{DW layers}) - \rho(\text{WZ}' \text{ layer}) - \rho(\text{DW layer})$. The isosurface charge density in the figure is $\sim 8.6 \times 10^{-5} \text{ e/Bohr}^3$. The electrostatic columbic interaction between the two CDWs with opposite charges lower down the energy, and thereby stabilize the CDW system.

To conclude, ZB' type α -In₂Se₃ is considered to exhibit a direct bandgap with minimal thickness dependence, as indicated by its prominent PL peaks. In contrast, the WZ' type α -In₂Se₃ is characterized as an indirect semiconductor with a bandgap that shows a distinct dependence on thickness, and it does not exhibit PL peaks. These distinct characteristics serve as useful discriminators between the two phases.

Action: We have added these thickness dependent optical properties and the related discussion into the revised manuscript to clarify the different bandgaps of WZ' type and ZB' type In₂Se₃.

5. *The authors should provide a modified version of figure 2b, of the HRSTEM data. They should clearly show how the quintuple layers would stack with an interlayer vdW gap to create the bulk heterostructure. It appears in the schematic of the suggested atom positions that the upper-most atoms of the quintuple layer have been placed so that there is not a significant gap between the quintuple layers. This is in contrast to the structures presented in this article: <https://pubs.acs.org/doi/10.1021/acs.inorgchem.8b01950>, where there is a significant vdW space gap between the layers.*

Our Reply: Thank you. As we respond to reviewer #1, we now provide a clearer HRSTEM image, as shown in **Fig. R5a**, with clear van der Waals gaps of ~ 1 Å. As indicated by the numbers, there are five stacking layers with varied atomic configurations. Layers 1, 3, and 5 exhibit the characteristic atomic configuration of the WZ' phase, each with opposite out-of-plane polarizations denoted by black arrows. These ferroelectric domains are separated by in-plane domain walls (IP-DWs) of varying types: layer 2 represents a head-to-head IP-DW, while layer 4 is a tail-to-tail IP-DW. These IP-DWs display a uniform non-polar state, with the central Se atoms positioned in the center of two neighboring In layers. This arrangement marks a sharp flip of the polarization vector direction, with the IP-DW width measuring a single unit cell along the c-axis (~ 0.8 nm) — even narrower than that observed in conventional three-dimensional ferroelectrics like BiFeO₃ (~ 2 nm; *Nat. Mater.* **8**, 229, 2009). Furthermore, we conducted a statistical analysis of the off-center displacements of the central Se atoms within the domain to evaluate the magnitude of spontaneous polarization. The displacement, measured as the distance from the central Se atoms to their symmetric position along the c-axis, is ~ 0.3 - 0.4 Å as illustrated in **Fig. R5b**, which aligns well with ~ 0.4 Å derived from theoretical calculations.

Fig. R5. Atomically resolved characterization of the ferroelectric domain configuration in WZ'-type α -In₂Se₃. **a.** Five typical layers with van der Waals gaps of ~ 1 Å are clearly seen. Each layer comprises five atoms arranged in an atomic sequence of Se-In-Se-In-Se, which are bounded by covalent bonds. WZ' phase of different polarization directions (layer 1, 3, and 5) are isolated by In-plane domain walls (IP DW, layer 2 and 4). Their atomic microstructures are schematically shown in typical regions marked by rectangles of different colors. **b.** The displacement of the central Se atom in each layer along the c axis. 10-unit cells have been counted in each layer. Inset schematically shows the IP-DWs and out-of-plane polarizations.

6. The authors describe the ferroelectric properties of the grown material. They appear to attribute most of the strong ferroelectric effect to trapped charges rather than the intrinsic

properties of the material itself. Given that the intrinsic ferroelectric effect is not as important, the authors should expand on whether, for this type of mechanism, the WZ phase is a more effective material than the ZB for these types of devices.

Our Reply: Thank you for your question. As illustrated in **Figs. 4e & 4f**, the current switching arises from the combined effects of ferroelectric polarization switching and charge trapping/de-trapping processes. A larger ferroelectric polarization will promote more extensive charge trapping/de-trapping during polarization reversal. Following your suggestion, we fabricated an in-plane Pt/ZB'-In₂Se₃/Pt device, with the ZB'-In₂Se₃ film deposited on a mica substrate (schematic inset in **Fig. R16**). The *I-V* curves measured at various frequencies reveal only minor hysteresis, with no observable current switching peaks—a stark contrast to the behavior of WZ'-phase-based devices (**Fig. 4a** in the manuscript). This difference likely stems from the weaker in-plane (IP) ferroelectric polarization of the ZB' phase compared to the WZ' phase. We note that the modest resistive switching in ZB'-phase devices could be enhanced through interface engineering. For instance, inserting a thin SiO₂ layer between ZB'-In₂Se₃ and Au electrodes (e.g., Au/SiO₂/ZB'-In₂Se₃/SiO₂/Au), as recently demonstrated by Y.-R. Jeon et al. (*Adv. Mater.* **2025**, 37, 2413178), may improve the device performance. This suggests that both phase structure and interfacial design are critical for optimizing ferroelectric synaptic properties.

We have included **Fig. R16** and the related discussion into the revised manuscript.

Fig. R16. *I-V* characteristics of two terminal devices based on ZB' type α -In₂Se₃ films. Various frequencies ranging from 0.008 Hz to 0.044 Hz have been used. The up-inset shows the typical *I-V* curves in semi-log scale. The down-inset schematically shows the device structure and measurement configuration. Minor current switching peaks have been observed.

Response to reviewers' comments

Dear Reviewers,

Thank you very much for your thorough review and critical comments, which help us improve the manuscript significantly. Below we provide a point-to-point response to address your additional concerns. We note that your comments are in *blue italic*, words in black font are our responses, and changes made to the manuscript are in red.

Reviewer #1:

The authors have well addressed my concerns and revised the manuscript accordingly. I would recommend the manuscript for publication.

Our Reply: Thank you for recommending the publication of our work in Nature Communications.

Reviewer #2:

I am satisfied that the authors have answered my initial questions on their manuscript very thoroughly. They have undertaken substantial new work and detailed further analysis which makes their new conclusions more convincing. The new HRTEM figures with supporting DFT calculations are illuminating and, along with the band-gap calculation, support the interesting proposal of the emergence of charged domain walls. I recommend that this article can be published after the authors have considered the additional points below.

Our Reply: Thank you very much for your constructive comments, and for recommending the publication of our work in Nature Communications. In the following, we address the additional points you raised.

- I would suggest that the authors state the quantitative difference in formation energies of the different phases discussed on line 291.

Our Reply: Yes. We have stated the quantitative energy values in the revised manuscript as follows, “And the long-range attractive Coulombic interactions between neighbouring CDW with opposite charges can reduce the total energy from ~ -3.632 eV/atom to ~ -3.636 eV/atom, making it energetically stable when compared to WZ'-In₂Se₃ with NDW (~ -3.634 eV/atom).”

- For full clarity, the authors should state the nature of the vdW correction scheme used in the calculations.

Our Reply: Thank you for your suggestion. We have included additional details on the vdW correction scheme into the “**Electronic band structure calculations**” section of

Methods. The revised text now states: “Given the critical influence of van der Waals (vdW) interactions, interlayer vdW forces were appropriately included **using the widely applied DFT-D3 empirical correction as implemented in VASP⁶⁶**. **In this vdW correction scheme, the atom-pairwise specific dispersion coefficients and cutoff radii are both computed from first principles.**” Additionally, reference 66 has been updated to a more appropriate citation.

[66] Grimme, S. *et al.* A consistent and accurate ab initio parametrization of density functional dispersion correction (DFT-D) for the 94 elements H-Pu. *The Journal of Chemical Physics* **132** (15), 154104 (2010).

- I believe that the authors should be careful about statements made about the band gap of the Wurtzite type being 0.8 eV. Their own DFT calculations suggest that the bulk Wurtzite type in its simplest form has a band gap of 1.37 eV. It is only with the consideration of the formation of domain walls (and a hybrid structure is used within the DFT calculations) when a band gap of 0.8 eV is obtained. Are the authors proposing that the Wurtzite phase will always form with the proposed domain structure, or is this a result of the growth conditions of this work? Perhaps a sentence such as "the band gap can be tuned to"... or similar would be appropriate? I would also be careful about the statement in the abstract about this point, where for clarity the domain walls should be mentioned.

Our Reply: Thank you. You’re right that the formation of domain walls reduces the bandgap from ~ 1.37 eV to 0.8 eV in bulk Wurtzite-type In_2Se_3 . While domain structures naturally occur in various ferroelectric materials, we are not proposing that WZ’- In_2Se_3 will always form with the proposed domain structure, it is the result of the growth conditions in this work. Other domain wall types may exist, and their potential influence on the bandgap remains an open question for future investigation. To clarify this point, we have modified the manuscript to describe the bandgap as tunable. Key revisions include: (1) In the abstract, “We demonstrate that it is a narrow bandgap ferroelectric semiconductor, featuring a Curie temperature exceeding 620 K, **a tunable bandgap (0.8 – 1.6 eV) modulated by charged domain walls**, and a large optical absorption coefficient of $1.3 \times 10^6/\text{cm}$.”. (2) In the summary, “This marks the inaugural experimental preparation of WZ’ type $\alpha\text{-In}_2\text{Se}_3$, confirmed as a 2D FE semiconductor with a high T_c exceeding 620 K **and a tunable bandgap of down to 0.8 eV. The bandgap exhibits a strong reduction from 1.6 eV to 0.8 eV as the film thickness increases from 3 nm to 25 nm, primarily attributed to the charged domain walls within the film.**”